# An HSV-1-H129 amplicon tracer system for rapid and efficient monosynaptic anterograde neural circuit tracing

Feng Xiong[1,2,3,12], Hong Yang[1,2,12], Yi-Ge Song[4], Hai-Bin Qin[1,2], Qing-Yang Zhang[1,2], Xian Huang ⬡[4], Wei Jing[4], Manfei Deng[4], Yang Liu[5], Zhixiang Liu[5], Yin Shen ⬡[6], Yunyun Han ⬡[6], Youming Lu ⬡[4], Xiangmin Xu[7,8], Todd C. Holmes ⬡[8,9], Minmin Luo ⬡[5,10], Fei Zhao ⬡[10,11] ✉, Min-Hua Luo ⬡[1,2,3,8] ✉ & Wen-Bo Zeng ⬡[1] ✉

Monosynaptic viral tracers are essential tools for dissecting neuronal connectomes and for targeted delivery of molecular sensors and effectors. Viral toxicity and complex multi-injection protocols are major limiting application barriers. To overcome these barriers, we developed an anterograde monosynaptic H129$_{Amp}$ tracer system based on HSV-1 strain H129. The H129$_{Amp}$ tracer system consists of two components: an H129-dTK-T2-$pac$^Flox helper which assists H129$_{Amp}$ tracer's propagation and transneuronal monosynaptic transmission. The shared viral features of tracer/helper allow for simultaneous single-injection and subsequent high expression efficiency from multiple-copy of expression cassettes in H129$_{Amp}$ tracer. These improvements of H129$_{Amp}$ tracer system shorten experiment duration from 28-day to 5-day for fast-bright monosynaptic tracing. The lack of toxic viral genes in the H129$_{Amp}$ tracer minimizes toxicity in postsynaptic neurons, thus offering the potential for functional anterograde mapping and long-term tracer delivery of genetic payloads. The H129$_{Amp}$ tracer system is a powerful tracing tool for revealing neuronal connectomes.

Retrograde monosynaptic tracers derived from rabies virus have been broadly applied for anatomical mapping of the input connectome of neural circuits and functional targeted delivery of genetically encoded sensors and effectors[1–3]. Although multiple anterograde monosynaptic tracers have been developed based on the herpes simplex virus 1 (HSV-1) strain H129 (H129), functional mapping for output connectome requires

further technical development to minimize viral toxicity. Retrograde and anterograde monosynaptic tracing can be achieved by combining two components: (i) a conditionally competent viral tracer (tracer), and (ii) a helper virus (helper). Replication- and/or transmission deficiencies due to the deletion of one or more genes that are required for transneuronal spread to connected neurons of the viral tracer component is a

[1]State Key Laboratory of Virology, CAS Center for Excellence in Brain Science and Intelligence Technology, Wuhan Institute of Virology, Chinese Academy of Sciences, Wuhan, China. [2]University of Chinese Academy of Sciences, Beijing, China. [3]Key Laboratory of Magnetic Resonance in Biological Systems, State Key Laboratory of Magnetic Resonance and Atomic and Molecular Physics, National Center for Magnetic Resonance in Wuhan, Wuhan Institute of Physics and Mathematics, Innovation Academy of Precision Measurement Science and Technology, Chinese Academy of Sciences, Wuhan, China. [4]Department of Physiology, School of Basic Medicine, Tongji Medical College, Huazhong University of Science and Technology, Wuhan, China. [5]National Institute of Biological Sciences, Beijing, China. [6]Eye Center, Renmin Hospital, Wuhan University, Wuhan, China. [7]Department of Anatomy and Neurobiology, School of Medicine, University of California, Irvine, CA, USA. [8]Center for Neural Circuit Mapping, School of Medicine, University of California, Irvine, CA, USA. [9]Department of Physiology and Biophysics, School of Medicine, University of California, Irvine, CA, USA. [10]Chinese Institute for Brain Research, Beijing, China. [11]School of Basic Medical Sciences, Capital Medical University, Beijing, China. [12]These authors contributed equally: Feng Xiong, Hong Yang. ✉e-mail: zhaofei@cibr.ac.cn; luomh@wh.iov.cn; zengwb@wh.iov.cn

feature that enforces monosynaptic spread. The helper virus component expresses genes that complement the tracer's deficient gene(s), thus supporting tracer replication and/or transmission[4–6]. To maintain the limit of monosynaptic spread, the helper virus must be limited to the initial set of infected cells. To date, all the published monosynaptic tracers use replication-deficient adeno-associated virus (AAV) as the helper virus component[1,6–9]. Due to the slow onset of expression for AAV vectors, the use of AAV necessitates two separate injections, first helper and then tracer, to allow sufficient expression and accumulation of the complementary gene and to support the tracer's replication and transmission. Subsequently, the tracer is injected into the same location as the previous AAV helper injection 2–3 weeks later[1,6–9]. Starting from helper injection to brain collection, the duration of these multi-step experiments typically takes 3–4 weeks. One obvious disadvantage of this protocol is that the obligatory sequential injections of the helper and tracer potentially leads to insufficient overlap of their injection locations due to injection variation of the sites of their respective injections, which can lead to variable results between experiments and inefficient tracing due to spatial mismatch of the helper and tracer injection sites.

Relative to progress on retrograde neural tracers, the development of useful anterograde tracers is far behind. To date, there are six published anterograde monosynaptic tracer systems that employ to three types of viruses. The first is transneuronal AAV1[10] which requires no helper and is thus an exception to the above-mentioned scheme for two component helper-dependent tracers[10,11]. AAV1 has several disadvantages for neuronal tracing: (i) it fails to perform starter-specific tracing; (ii) it requires high viral dosages which often leads to undesired retrograde labeling and unintended leak to nearby areas[4]; and (iii) it requires an additional reporter system (e.g., Cre- or flippase-dependent fluorescent protein expression) to amplify the signal to compensate for poor transneuronal transmission efficiency[10,11]. Furthermore, AAV1 can only carry small genetic payloads under 5 kb. These combined shortcomings limit the utility of transneuronal AAV1[10,12]. The second system is a newly developed tracer derived from YFV-17D, a live attenuated yellow fever vaccine virus, and the third is an AAV expressing mCherry fused with the improved wheat germ agglutinin (AAV2-mWGA-mCherry)[6,13]. Both YFV-17D derived and AAV2-mWGA-mCherry anterograde tracers can map monosynaptic projectomes in wild-type and Cre mice, and represent promising new systems that merit further investigation. The remaining anterograde tracing systems are all derived from H129, an HSV-1 strain that predominantly spreads anterogradely. The H129-based system has been modified in two ways to include TK deficient tracer versions (H129-dTK-tdT, H129-dTK-T2) and gK deficient tracer versions (H129-dgK-G4), representing improved tracing systems[7,9,14]. H129-dTK-tdT and H129-dTK-T2 exhibit limited labeling efficiencies, but H129-dgK-G4 exhibits higher labeling efficiency with improved anterograde specificity[7,9,14]. However, these H129 modified variants all confer strong neuronal toxicity, thus limiting their use for functional connectome mapping for the targeted delivery of sensors and effectors[9,14,15]. These toxicity issues are further complicated by requirements for sequential injection protocols and long experimental durations. These limiting features are shared by most current anterograde and retrograde monosynaptic transneuronal viral tracer systems[4]. Ongoing efforts focus on circumventing these limitations for the next generations of novel monosynaptic tracer systems.

The HSV-1 amplicon is a pseudovirus particle that carries a pseudo-genome, a concatameric form of a multiunit amplicon plasmid. The HSV-1 pseudovirus particle is equipped with the identical capsid, tegument layer, and envelope proteins as the wild-type HSV-1, thus sharing identical host/cell tropism and infection features[16]. The amplicon plasmid contains Ori and pac, two essential cis-elements of HSV-1 genome. Ori plays a critical role in initiating the amplicon plasmid replication and forming the concatameric pseudo-genome, and

pac is essential for the pseudo-genome recognition and packaging into the viral capsid[17,18]. The HSV-1 amplicon shows promise as a gene transfer vector based on its unique features: (i) broad cell tropism that mimics the infectivity of wild-type HSV-1; (ii) large transgene capacity that supports a maximum of ~150 kb genetic payload[19,20]; (iii) high transgene expression efficiency, expressing the target gene(s) simultaneously with the multiunit expression cassettes (multiple-copy of target); and (iv) vastly minimized toxicity as it neither replicates nor produces toxic viral proteins[16,20].

In the present study, we developed an anterograde monosynaptic tracer system by using H129 amplicon (H129$_{Amp}$). The H129$_{Amp}$ tracer system is composed of H129$_{Amp}$ tracer and H129-dTK-T2-pac$^{Flox}$ helper. H129-dTK-T2-pac$^{Flox}$ helper has the thymidine kinase gene deleted (dTK), expresses two copies of the fluorescent protein tdTomato (T2), and one copy of pac removed while the remaining one flanked by loxN (LoxN-pac-LoxN, pac$^{Flox}$) which can be subsequently excised by Cre-recombinase. The H129$_{Amp}$ tracer contains multiple-copy genetic payloads that express simultaneously, conferring high expression efficiency and high labeling intensity. We generated three different tracers, H129$_{Amp}$-CTG, H129$_{Amp}$-DIO-TG, and H129$_{Amp}$-Flp-DIO-TG to achieve a range of experimental goals. All three tracers utilize the same helper to produce/package the H129 amplicon ("tracer") to facilitate anterograde spread to postsynaptic neurons. The identical virion structure determines the commonly shared infection features of the tracer/helper and thus requires only a single injection instead of the conventional requirements for sequential injections. As a result, the H129$_{Amp}$ tracer system labels postsynaptic neurons with high labeling intensity within 5 days. Notably, the H129$_{Amp}$ tracer system displays minimized toxicity in the postsynaptic neurons since H129$_{Amp}$ tracers do not express toxic viral proteins. This feature of minimized postsynaptic toxicity offers considerable potential for functional mapping studies using optogenetic effectors combined with electrophysiological assays.

Input-defined postsynaptic neurons' monosynaptic anterograde tracing has been successfully acheived by combining H129$_{Amp}$ tracer system with an anterograde monosynaptic tracer H129-dgK-G4, which was introduced very recently[9]. H129$_{Amp}$ tracer system was simultaneously injected in a brain region of interest (1st order initially infected neurons). H129-dgK tracer pair (helper AAV2/9-DIO-mCh-gK and tracer H129-dgK-G4) were sequentially injected in the 2nd-order neuronal target sites that are innervated by the 1st-order neurons. The newly synthesized H129$_{Amp}$ tracer is transmitted to postsynaptic neurons, making them the input-defined 2nd-order neurons. There, the Cre-recombinase expressed by H129$_{Amp}$ tracer initiates the Cre-dependent AAV helper (AAV2/9-DIO-mCh-gK) expressing gK to support H129-dgK-G4 further anterogradely transmission to the 3rd-order neurons, thus achieving input-defined postsynaptic neurons' monosynaptic anterograde tracing by labeling the 3rd-order neurons.

The H129$_{Amp}$ tracer system represents a significant advance for anterograde monosynaptic tracing, and its applications will contribute to revealing the output connectome by combining anatomical tracing with targeted functional circuit analysis.

## Results

### Generation, production, and tracing principles of the H129$_{Amp}$ tracer system

Ori and pac are two essential cis-elements for replication and packaging of the HSV-1 genome[21–24]. We obtained the amplicon plasmid backbone pHSV by cloning Ori and pac from H129 to the pCDNA3.0 plasmid. Subsequently, an expression cassette of Cre-recombinase (Cre), HSV thymidine kinase (TK), and GFP were inserted into the pHSV construct, generating the amplicon plasmid pHSV-Cre-TK-GFP (pHSV-CTG), the basic unit of H129$_{Amp}$ tracer's pseudo-genome (Fig. 1a). The helper is derived from H129-dTK-T2, the TK deficient, 2 × tdTomato expressing recombinant virus introduced previously[14]. Briefly, for the

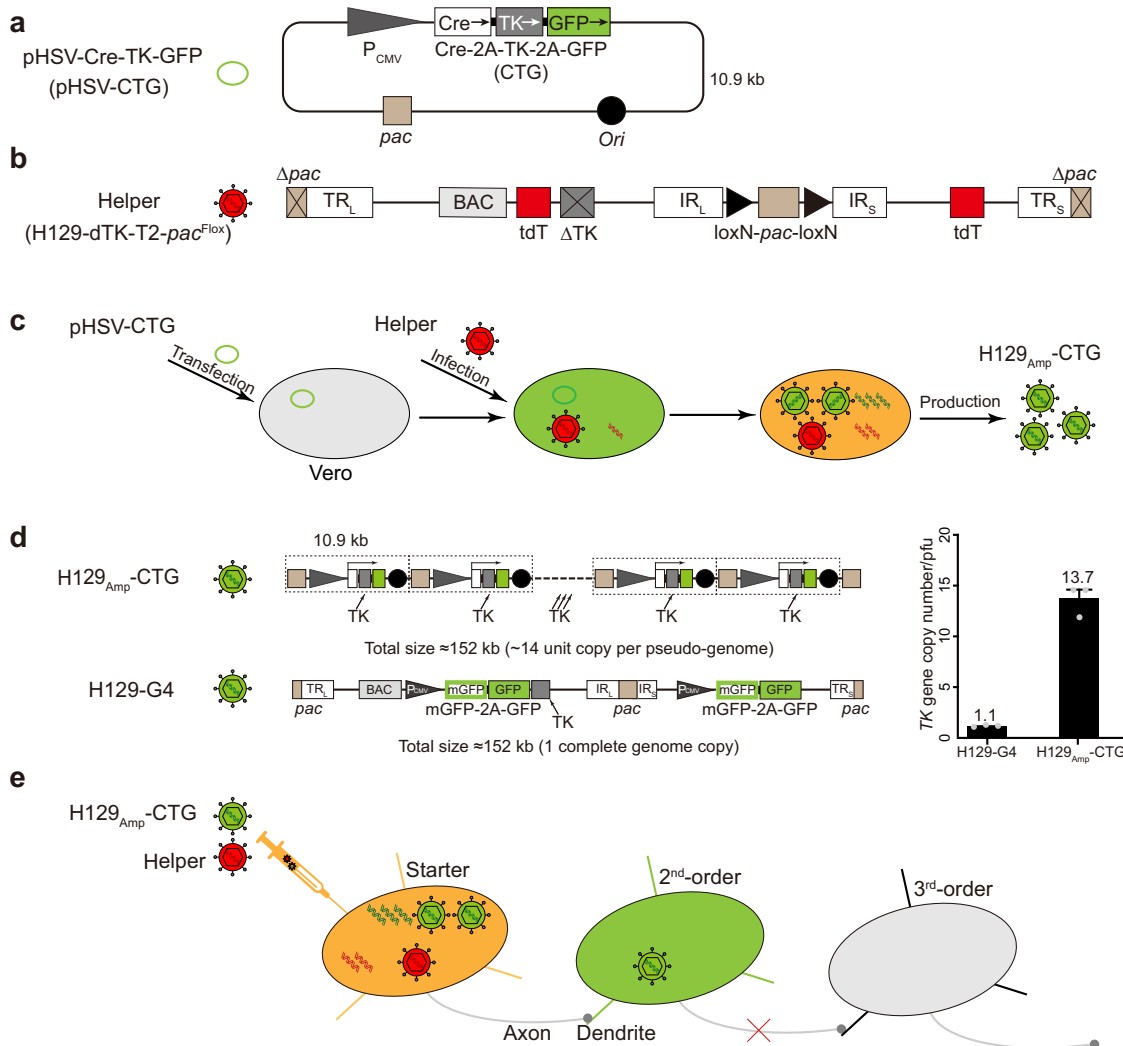

**Fig. 1 | H129 amplicon-derived anterograde monosynaptic tracer system: H129$_{Amp}$-CTG tracer and H129-dTK-T2-*pac*$^{Flox}$ helper. a** Schematic structure of amplicon plasmid pHSV-Cre-TK-GFP (pHSV-CTG). *Ori* and *pac* of H129 were cloned into pcDNA3.0 to generate the backbone pHSV. Then CMV-promoter-controlled expression cassette of Cre, HSV-TK, and GFP was inserted into pHSV, generating pHSV-CTG. **b** Schematic genome structure of H129-dTK-T2-*pac*$^{Flox}$ helper. H129-dTK-T2-*pac*$^{Flox}$ carries a viral genome with the TK deletion (dTK), two tdTomato (tdT) expression cassettes (T2), one *pac* removed (Δpac), and the remaining *pac* flanked by LoxN sequences (LoxN-pac-LoxN, *pac*$^{Flox}$), which can be excised by Cre-recombinase. **c** Production of H129$_{Amp}$ tracer. Vero cells are transfected with pHSV-CTG for 24 h, then infected with H129-dTK-T2-*pac*$^{Flox}$. The helper provides all the viral proteins for amplicon replication and packaging. Cre expressed by H129$_{Amp}$-CTG excises the floxed-*pac* in H129-dTK-T2-*pac*$^{Flox}$ genome, disarms its genome packaging to produce helper virus. The replicated pHSV-CTG, containing *pac* signal, is packaged into viral particle as a pseudo-genome, generates the H129$_{Amp}$-CTG tracer. **d** Schematic genome structure of the representative amplicon tracer H129$_{Amp}$-CTG. The amplicon pseudo-genome, a similar size to the wild-type H129 genome (~152 kb), contains multiunit pHSV-CTG sequences (~14 units) (upper panel). While H129-G4 contains only 1-copy *TK* gene (lower panel). The absolute copy numbers of *TK* gene in the H129$_{Amp}$-CTG pseudo-genome and H129-G4 genome were determined by quantitative PCR, then normalized to the corresponding virus titer determined by plaque-forming assay. The experiment was performed in triplicate, results are presented as means ± SEM from three independent experiments. The results indicate that H129$_{Amp}$-CTG contains 13.7 ± 0.90 (means ± SEM) copies and H129-G4 contains only 1.1 ± 0.05 (means ± SEM) copies of *TK* gene/pfu on average (right panel). **e** The schematic mechanism for monosynaptic tracing with H129$_{Amp}$-CTG tracer system. Neither H129$_{Amp}$-CTG nor H129-dTK-T2-*pac*$^{Flox}$ propagates alone in neurons. When co-infecting the same neuron, H129-dTK-T2-*pac*$^{Flox}$ provides all the necessary viral proteins to support H129$_{Amp}$-CTG replication and packaging, while its own encapsidation is disarmed since the floxed-*pac* is excised by Cre expressed from H129$_{Amp}$-CTG. The newly produced H129$_{Amp}$-CTG tracer, with the identical viral particle structure and transportation properties with wild-type H129, anterogradely transmits to postsynaptic neurons, and massively expresses GFP labeling these cells. H129$_{Amp}$-CTG is restrained in the 2nd-order cells without further spread due to the lack of helper. Source data are provided as a Source Data file.

construction of H129-dTK-T2-*pac*$^{Flox}$, the *pac* sequences located in the terminal repeats (TRs) were removed from the bacterial artificial chromosome (BAC) of H129-dTK-T2 (H129-dTK-T2-BAC). The remaining complete *pac* sequence was flanked by loxN (loxN-*pac*-loxN, *pac*$^{Flox}$) and was reconstituted as the recombinant helper virus H129-dTK-T2-*pac*$^{Flox}$ (helper) (Fig. 1b).

H129$_{Amp}$ tracer, represented by H129$_{Amp}$-CTG, was produced in Vero cells as described in the Methods. Briefly, Vero cells were initially transfected with pHSV-CTG, and then infected with the helper, which

was separately propagated in Vero cells as described[7]. TK deficiency severely impairs helper replication in neurons, but not in Vero cells which fully support viral genome replication, viral protein synthesis, as well as production of helper[25]. In Vero cells, the helper supports the production of high-titer H129$_{Amp}$ tracer, pseudovirus particles, by providing all viral functional and structural proteins necessary for the replication and packaging of pseudo-genome of concatemeric pHSV-CTG[18]. H129$_{Amp}$-CTG expresses Cre which efficiently excises the floxed-*pac* in the H129-dTK-T2-*pac*$^{Flox}$ genome. Loss of *pac* disarms

encapsidation of H129-dTK-T2-*pac*^Flox genome, thus terminates helper production[18] (Fig. 1c). While the recombination efficiency of the Cre-recombinase appears to be high, a very small fraction of the floxed-*pac* in replicated helper genomes remains unexcised, thus resulting in normal genome packaging and a trace amount of helper production. Therefore, the raw H129$_{Amp}$ product made in Vero cells is a mixture of H129$_{Amp}$ tracer (~95%) and H129-dTK-T2-*pac*^Flox helper (~5%). These titers of H129$_{Amp}$ tracer, as well as helper "contamination", in the raw H129$_{Amp}$ product were determined by plaque-forming assays and judged by their respective fluorescences[14,26]. Independently propagated helper was added to the tracer raw product to adjust the tracer and helper to the working titers, which have been carefully optimized for the best tracing efficiency (Supplementary Fig. 1). The titer optimized tracer/helper mixture was preserved at −80 °C as aliquots, designated as H129$_{Amp}$ tracer system.

Although the H129$_{Amp}$ tracer efficiently expresses Cre, TK, and GFP from the concatemeric multiunit cassettes in the amplicon pseudo-genome (represented by H129$_{Amp}$-CTG as shown in Fig. 1d)[27], the H129$_{Amp}$ tracer by itself does not replicate in, nor does it spread between neurons, as it contains no viral genes other than *TK*. This was confirmed by injecting the raw product of H129$_{Amp}$-CTG tracer in the absence of the helper, designated as "tracer alone", although a very small fraction of helper "contamination"(5%) is inevitable and cannot be removed in the raw H129$_{Amp}$ tracer product (Supplementary Fig. 2). Conversely, the H129-dTK-T2-*pac*^Flox helper does not spread by itself in the absence of the tracer (Supplementary Fig. 3), because the lack of TK severely impairs its replication in neurons without tracer co-infection[28]. Anterograde monosynaptic tracing occurs only when the tracer (represented by H129$_{Amp}$-CTG) and the helper co-infect the same neurons when are simultaneously administrated into the brain area of interest. These viral tracer components do not distinguish between outputs of different neuron types in the brain area with the optimized dosage and ratio of the tracer/helper co-infected starter neurons[19,29] (Fig. 1e). In co-infected starter neurons, H129$_{Amp}$-CTG tracer expresses TK to assist helper genome replicating and expressing all required viral proteins, which further support H129$_{Amp}$ tracer replication and packaging. The newly produced H129$_{Amp}$-CTG tracer then transmits through the axon and infects connected postsynaptic neurons. There, GFP is rapidly and massively expressed from the tracer's pseudo-genome and labels postsynaptic neurons. Floxed-*pac* is removed from the helper genome by Cre-excision, the helper is not packaged and will not spread beyond the initially infected neurons, since *pac* is required for viral protein recognizing and packaging the helper genome to produce helper. Similarly, the H129$_{Amp}$-CTG tracer is also unable to spread beyond initially infected neurons because it is unable to replicate in the absence of helper.

## Anterograde monosynaptic tracing with H129$_{Amp}$ tracer systems: fast-bright tracing

To determine tracing performance, the H129$_{Amp}$ tracer system (H129$_{Amp}$-CTG tracer and helper H129-dTK-T2-*pac*^Flox) was injected into the primary auditory cortex (AC) of wild-type C57BL/6 mice (Fig. 2a, b). The tracing results are observed at 5 days post-injection (Day 5). This timepoint was determined empirically as the optimized observation timepoint for monosynaptic tracing (Supplementary Fig. 4). Abundant AC neurons are labeled, of which 51% are co-infected by both tracer and helper (Fig. 2c, d, yellow). These dual labeled neurons represent the potential starter neurons. Other singly infected neurons are infected by either H129$_{Amp}$ tracer alone (green, 31%) or by helper alone (red, 18%). These singly infected neurons are unable to spread further tracing (Fig. 2c, d).

Many GFP^+ neurons are clearly observed in downstream nuclei, which are directly innervated by AC neurons. Labeled postsynaptic neurons include those in contralateral auditory cortex (Cont. AC), lateral amygdaloid nucleus (LA), medial geniculate nucleus (MG),

external globus pallidus (GPe) and locus coeruleus (LC) (Fig. 2e–i). Due to Cre-recombination efficiency, the helper has a low probability to be packaged with unexcised *pac*, and transmitted to the postsynaptic neuron, which may potentially result in multi-synaptic labeling. But neither helper labeled neurons are detected in the AC-innervating regions (Fig. 2 and Supplementary Fig. 4), nor are tracer labeled neurons detected in the further downstream regions (Supplementary Fig. 5) in our experiments. These results indicate that H129$_{Amp}$ tracer anterogradely selectively labels directly connected postsynaptic neurons.

Anterograde monosynaptic tracing with H129$_{Amp}$ tracer system was further confirmed in Ai14 reporter mice. In these mice, Cre expressed by H129$_{Amp}$-CTG efficiently drives long-term and robust expression of the Cre-dependent fluorescence reporter. The results show a prolonged tracing observation time window of at least 14 days (Supplementary Fig. 6). Despite the predominant anterograde transneuronal transmission, H129-derived tracers have the potential to retrogradely label the upstream neurons by invading the axon terminal and retrograde transportation[30,31]. However, under these experimental conditions that have been carefully optimized (dose, ratio, volume, timing, etc), no labeled cell bodies in the upstream region are detected, thus indicating the absence of retrograde labeling by the H129$_{Amp}$ tracer system (Supplementary Fig. 6). Proper controls are still strongly recommended for each application when applying the H129$_{Amp}$ tracer system to optimize the experiment parameters so that potential retrograde labeling can be ruled out.To perform starter-specific monosynaptic tracing, we generated an amplicon plasmid pHSV-DIO-TK-GFP (pHSV-DIO-TG) to produce the tracer H129$_{Amp}$-DIO-TG. The same helper H129-dTK-T2-*pac*^Flox is used in this starter-specific tracer system, but the tracer H129$_{Amp}$-DIO-TG expresses TK and GFP (without Cre) in a Cre-dependent manner (Supplementary Fig. 7a, b). After being adjusted to optimized titer (1.5 × 10^8 pfu/ml for each) and ratio (1:1), the H129$_{Amp}$-DIO-TG tracer and H129-dTK-T2-*pac*^Flox helper (H129$_{Amp}$-DIO-TG tracer system) were co-injected into the AC of Rph3a-Cre transgenic mice (Supplementary Fig. 7c, d). Rph3a-Cre transgenic mice were created by inserting P2A-Cre between Exon 22 and 3′-UTR region of *Rph3a* gene (unpublished data). At Day 5, H129$_{Amp}$-DIO-TG tracer and helper co-labeled neurons (merged as yellow) are observed at the injection site, representing potential starter neurons (Supplementary Fig. 7e). The neurons of direct downstream nuclei of AC, represented by the MG, Cont. AC, and LA, are labeled by H129$_{Amp}$-DIO-TG tracer (green neurons, Supplementary Fig. 7e–g). These results show that the H129$_{Amp}$-DIO-TG tracer system is also capable of performing output tracing from a specific neuron subpopulation.

The H129$_{Amp}$ tracer systems successfully achieves fast-bright anterograde monosynaptic tracing by a single-injection protocol within 5 days post-injection. This is a marked improvement compared to conventional anterograde tracing systems that require sequential injections and longer (4 weeks) experimental durations for mono-synaptic tracing.

## Anatomical and functional outputs mapping of the input-defined neurons with H129$_{Amp}$ tracer systems

Next, we mapped the outputs of the input-defined neuron sub-population anatomically and functionally using the H129$_{Amp}$ tracer system. H129$_{Amp}$ tracer system (H129$_{Amp}$-CTG tracer and helper) was administrated into the AC in the left hemisphere (L-AC) of wild-type C57BL/6 mice as described above, and AAV2/9-DIO-ChR2-mCh was simultaneously injected into the right hemisphere AC (R-AC) in the same mice (Fig. 3a). R-AC receives inputs from the L-AC, and projects back to L-AC, as well as the right hemisphere MG and LA (R-MG and R-LA) (Fig. 3a). The newly propagated H129$_{Amp}$ tracer transmits from the L-AC to the postsynaptic neurons in the R-AC. This provides Cre which drives AAV2/9-DIO-ChR2-mCh to express ChR2-mCherry. As

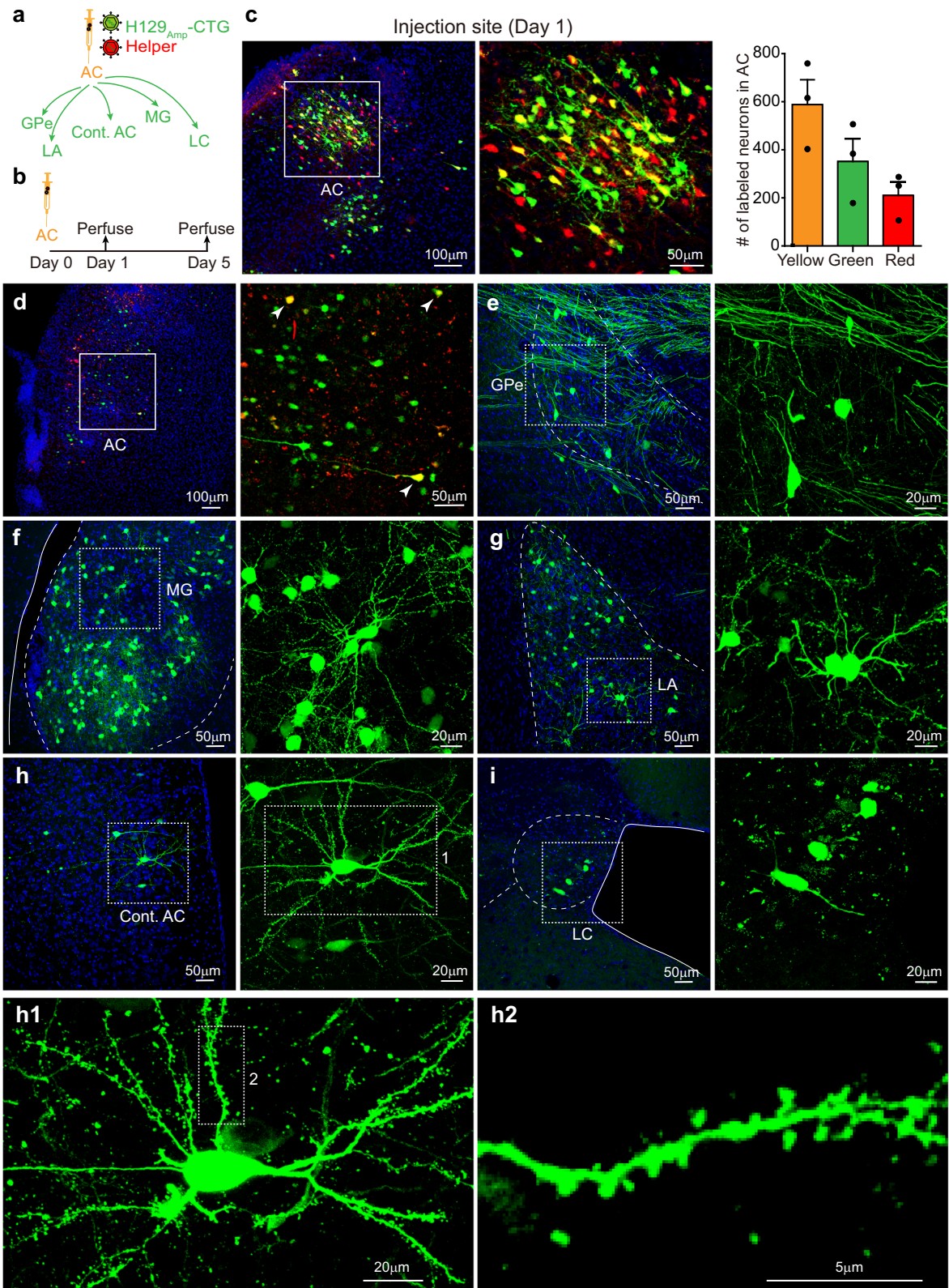

we expected, on Day 21 the soma of mCherry labeled neurons (mCh[+]) are clearly observed in the R-AC (Fig. 3b), but not in any other regions (data not shown). Notably, mCherry labeled axonal fibers, belonging to the labeled R-AC neurons, are also observed in the L-AC, R-MG, and R-LA (Fig. 3c–e), but not in the L-MG and L-LA (data not shown). Since the fluorescent protein expressed by either H129$_{Amp}$ tracer or helper is no longer detectable after Day 7 (Supplementary Fig. 4), mCherry from

AAV2/9-DIO-ChR2-mCh is the only source for red fluorescence signal observed at Day 21.

H129$_{Amp}$ tracers express only transgenes, but no toxic viral proteins. This results in no or minimized toxicity after transmission to the postsynaptic neurons[32]. At Day 21, mCherry expressing (mCh[+]) R-AC neurons (the H129$_{Amp}$ tracer labeled postsynaptic neurons) maintain apparently normal cell morphology (Fig. 3b). Moreover, these neurons

**Fig. 2 | Fast-bright tracing the auditory cortex projection pathways with H129$_{Amp}$ tracer system. a** Schema of the simplified projection pathways of the auditory cortex. AC auditory cortex, Cont. AC contralateral AC, MG medial geniculate nucleus, LA lateral amygdala, GPe external globus pallidus, LC locus coeruleus. **b** Tracing AC outputs with H129$_{Amp}$ tracer system in wild-type C57BL/6 mice. The H129$_{Amp}$- tracer system (H129$_{Amp}$-CTG $1.5 \times 10^8$ pfu/ml and helper $1.5 \times 10^8$ pfu/ml, in 300 nl) was injected into the AC (AP: −2.80 mm; ML: −4.13 mm; DV: −2.38 mm) of wild-type C57BL/6 mice. The brains were collected at 1 and 5 days post-injection (Day 1 and Day 5), and images were obtained after cryosection and DAPI counterstaining. **c** Labeled neurons in the injection site at Day 1. Representative images of the injection site AC at Day 1 are shown (left panel), and the boxed region is displayed with higher magnification (middle panel). The numbers of AC neurons doubled-labeled by tdT/GFP (Yellow, 588 ± 103), or single-labeled by GFP (Green, 352 ± 95) or tdT (red, 210 ± 56) were quantified and presented as means ± SEM from three mice (right panel). **d−i** Representative tracing results at Day 5. Representative images of the injection site AC (**d**), and representative AC-innervating regions (**e−i**) are shown. Images with higher magnifications of the boxed areas are presented in the right panels. Potential starter neurons labeled by both GFP and tdTomato (merged as yellow) are indicated with white arrowheads. A representative GFP-labeled neuron in Cont. AC is further magnified and the morphological details are displayed (**h1** and **h2**). Source data are provided as a Source Data file.

display no apparent differences in membrane excitability as compared to adjacent non-infected (mCh⁻) neurons (Fig. 3f, g). Patch-clamp analysis of postsynaptic neurons labeled by the H129$_{Amp}$ tracer system indicates that these cells maintain normal physiological conditions and may be thus useful for functional connection mapping. However at present, we cannot rule out possible secondary effects of cell stress or immune responses that may impact yet to be determined aspects of cellular physiology.

To test further the functional output mapping potential of H129$_{Amp}$ tracer system, we prepared the acute brain slices containing ChR2-mCh⁺ axons in the L-AC, and recorded light-evoking EPSCs in L-AC neurons (Fig. 3h, left panel). Among the recorded L-AC neurons, about half (5/11) show fast light-evoked excitatory synaptic responses (Fig. 3h, middle and right panels). These results show that the H129$_{Amp}$ tracer system is compatible with functional assays, and has potential for use in functional mapping applications.

Notably, H129$_{Amp}$ tracer system is also applicable for input-defined output mapping from specific starter neurons. We modified the tracer by adding a flippase recombinase (Flp) expression cassette to H129$_{Amp}$-DIO-TG, generated H129$_{Amp}$-Flp-DIO-TG, which expresses Flp constitutively but expresses TK and GFP restrictively in a Cre-dependent manner (Supplementary Fig. 8a, b). The H129$_{Amp}$ tracer system (H129$_{Amp}$-Flp-DIO-TG tracer and helper) was injected into the inferior colliculus (IC) of CaMK2a-Cre transgenic mice that express Cre in glutamatergic neurons. The Flp-dependent reporter AAV2/9-fDIO-ChR2-mCh was simultaneously injected into the sub-parafascicular thalamic nucleus (SPF), a brain region directly innervated by IC that projects to the lateral superior olive (LSO) (Supplementary Fig. 8c, d). At Day 21, many SPF neurons (postsynaptic neurons of the IC excitatory neurons) are labeled with mCherry (Supplementary Fig. 8e). mCherry labeled axonal fibers projecting from the labeled SPF neurons are observed in LSO, showing further anatomical details of the SPF to LSO projection (Supplementary Fig. 8f).

Altogether, these results demonstrate that the H129$_{Amp}$ tracer systems combined with appropriate reporter AAVs may be used to map the output pathways of input-defined neuron subpopulations both anatomically and functionally (ChR2-assisted).

## Input-defined postsynaptic neurons' anterograde monosynaptic tracing by combination of H129$_{Amp}$ and H129-dgK-G4 tracer systems labels 3rd-order neurons

Output mapping of the input-defined neurons' postsynaptic neurons is critical for dissecting neuronal connectomes. While output mapping of the input-defined neurons has been performed only by visualizing the axonal fibers without identifying 3rd-order neurons thus far[10,12], in the present study, we combined H129$_{Amp}$ tracer system with an earlier developed anterograde monosynaptic tracer H129-dgK-G4[9], and achieved input-defined postsynaptic neurons' anterograde monosynaptic tracing. H129-dgK-G4 is a GFP-expressing recombinant H129 virus with deletion of the envelope protein glycoprotein K (gK)[9]. Cre-dependent AAV helper (AAV2/9-DIO-mCh-gK) complementarily restores gK in trans in the presence of Cre-recombinase, thus allowing H129-dgK-G4 tracer to anterogradely transmit to and label postsynaptic neurons from Cre-expressing starter neurons[9].

The H129$_{Amp}$ tracer system (H129$_{Amp}$-CTG tracer and helper) was administrated in a single injection into the L-AC of wild-type C57BL/6 mice at Day 0. AAV2/9-DIO-mCh-gK and H129-dgK-G4 were injected into R-AC of the same mice by sequential injections at Day 0 and 21, respectively (Fig. 4a). The scheme for the input-defined postsynaptic neurons' anterograde monosynaptic tracing includes the following steps: (i) the helper assists H129$_{Amp}$ tracer replication in the L-AC (1st-order neurons); (ii) newly produced H129$_{Amp}$ tracer anterogradely transmits through the first synapse to the postsynaptic neurons (2nd-order neurons), including R-AC neurons; (iii) in the R-AC, H129$_{Amp}$ tracer expresses Cre and drives AAV2/9-DIO-mCh-gK to express mCherry and gK, which in turn assists H129-dgK-G4 transmission; and (iv) the newly produced H129-dgK-G4 anterogradely transmits from the R-AC through the second synapse to label neurons with GFP in further downstream regions (3rd-order neurons), shown by labeling in L-AC, R-MG, and R-LA neurons (Fig. 4a). Notably, the fluorescence labeling from the H129$_{Amp}$ tracer system is no longer visible at Day 26, since the fluorescence signal from H129$_{Amp}$ tracer system diminished by Day 7 as described above (Supplementary Fig. 4). All observed fluorescence signals are from AAV2/9-DIO-mCh-gK and H129-dgK-G4 with Cre assistance from H129$_{Amp}$-CTG tracer at Day 26.

The tracing results obtained at Day 26 are consistent with our expectation that mCh⁺ neurons are observed in the R-AC neurons (2nd order). mCh⁺ neurons are L-AC-innervating neurons, which were co-infected with AAV2/9-DIO-mCh-gK (Cre-dependant mCherry and gK expressing) and H129$_{Amp}$ tracer (expressing Cre) transneuronally transmitted from L-AC. GFP⁺ neurons are observed in the R-AC, indicating H129-dgK-G4 infection and labeling. Some neurons are co-labeled with mCherry from AAV2/9-DIO-mCh-gK and GFP from H129-dgK-G4 (merged as yellow). These represent the potential starter neurons (2nd order) of defined inputs from L-RC (Fig. 4b). Notably, in the 3ʳᵈ-order nuclei, including L-AC, R-MG, and R-LA, many neurons are labeled with GFP from transmitted H129-dgK-G4. These neurons are the output targets of the R-AC neurons innervated by L-AC (Fig. 4c−e). In the absence of any one of the tracing components, input-defined postsynaptic neurons' anterograde monosynaptic tracing cannot be achieved, and no 3rd-order neurons are detected (Supplementary Fig. 9).

These results show that the H129$_{Amp}$ tracer system is capable of input-defined postsynaptic neurons' anterograde monosynaptic tracing when combined with H129-dgK-G4 monosynaptic tracer. Although we have not yet tested other combinations, starter-specific input-defined postsynaptic neurons' anterograde monosynaptic tracing could likely be achieved also using the modified amplicon tracer system (H129$_{Amp}$-Flp-DIO-TG and helper) together with the Flp-dependent H129-dgK tracing set (H129-dgK-G4 and AAV2/9-fDIO-mCh-gK), or Flp-dependent H129$_{Amp}$ tracer system (H129$_{Amp}$-Flp-TK-GFP and H129-dTK-T2-FRT-pac-FRT) together with the Cre and Flp-dependent H129-dgK tracing set (H129-dgK-G4 and AAV2/9-$_{Con}$/$_{Fon}$-mCh-gK) (Supplementary Fig. 10)

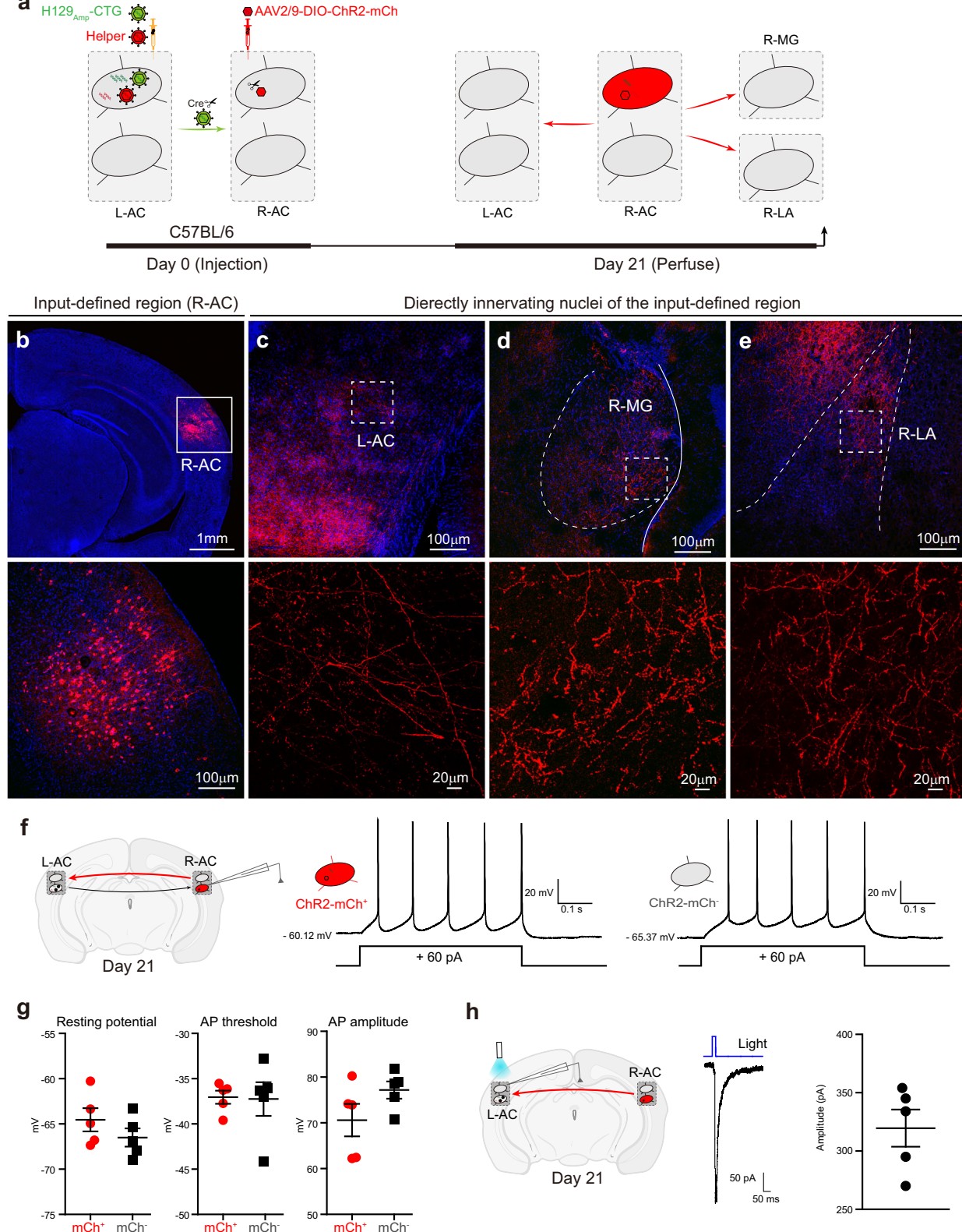

## Discussion

The HSV-1 amplicon has been considered as a promising gene delivery vector for decades based on its unique advantages, including broad cell tropism, very large capacity of genetic payload, minimized toxicity, high efficiency for target gene expression, and absence of genome integration features, etc[33]. However, the previously unremovable helper "contamination" in amplicon production greatly limits its applications.

In the present study, we developed an anterograde monosynaptic tracer system based on HSV amplicon, composed of the H129$_{Amp}$ tracer and H129-dTK-T2-$pac^{Flox}$ helper. This work innovatively applies the HSV-1 amplicon as neural circuit tracers, and converts the helper "contamination" in the raw amplicon product from a drawback for gene delivery to a favorable feature that facilitates monosynaptic tracing. Our work opens new useful applications of the HSV-1 amplicon

**Fig. 3 | Anatomical and functional outputs mapping of the input-defined neurons with H129$_{Amp}$ tracer system. a** Schematic illustration of the experiment setup and tracing strategy. The H129$_{Amp}$ tracer system (H129$_{Amp}$-CTG 1.5 × 10$^8$ pfu/ml and helper 1.5 × 10$^8$ pfu/ml, in 300 nl) was injected into the auditory cortex of the left hemisphere (L-AC, AP: −2.80 mm; ML: −4.13 mm; DV: −2.38 mm) of wild-type C57BL/6 mice; AAV2/9-DIO-ChR2-mCh (3.0 × 10$^{12}$ vg/ml, 100 nl), a reporter, was simultaneously injected into the R-AC (AP: −2.80 mm; ML: +4.13 mm; DV: −2.38 mm) of the same mice. The brains were collected at Day 21 for imaging or physiological assays. L-AC auditory cortex of the left hemisphere, R-AC AC of the right hemisphere, R-MG medial geniculate nucleus of the right hemisphere, R-LA lateral amygdala of the right hemisphere. **b–e** Representative tracing results. Shown are the representative images of input-defined region R-AC (**b**) and the R-AC-innervating regions (**c–e**). Images with higher magnifications of the boxed areas are presented in the lower panels. **f, g** Electrophysiological comparison of H129$_{Amp}$ labeled postsynaptic neurons and the adjacent non-labeled neurons. Current-clamp recordings were performed on the R-AC neurons to measure the electro-physiological parameters (**f**, left panel). Representative membrane responses of the H129$_{Amp}$ labeled postsynaptic R-AC neurons (mCh$^+$, **f**, middle panel) and the adjacent non-labeled normal neuron controls (mCh$^−$, **f**, right panel). Other electro-physiological parameters include resting potential (−64.54 ± 1.28 for mCh$^+$, −66.50 ± 1.01 for mCh$^−$), AP threshold (−37.05 ± 0.73 for mCh$^+$, −37.25 ± 1.86 for mCh$^−$) and AP amplitude (70.60 ± 3.56 for mCh$^+$, 77.17 ± 1.88 for mCh$^−$) expressed as means ± SEM (n = 10 from 3 mice) (**g**). AP, action potential. **h** Optogenetic connectivity mapping. Excitatory currents were recorded on the L-AC neurons of the brain slice at Day 21 with LED stimulation (n = 11 from 3 mice) (left panel). The excitatory currents (−70 mV) of a representative L-AC neuron (middle panel) and the mean amplitudes 319.6 ± 15.97 (means ± SEM) with LED-on are shown (right panel). Light stimulation duration is marked by the blue bar. Source data are provided as a Source Data file.

and creates an anterograde monosynaptic tracing tool with advantages of the simplified operation (single injection), shorter experimental duration (5 days), and greater potential for functional mapping, as well as input-defined postsynaptic neurons' anterograde monosynaptic tracing.

### The benefits of single injection

Monosynaptic viral tracer systems are usually composed of two components: (1) a replication- or transmission-defective viral tracer for transneuronal transmission/labeling, and (2) a non-replicable helper for supporting the defective tracer[4–6,9]. Conventionally, the helper (AAV) and the tracer are administrated by sequential injections. In the H129$_{Amp}$ tracer system, the recombinant self-replication-deficient helper virus not only functions as a helper for the H129$_{Amp}$ tracer production, but also works as a helper to support the H129$_{Amp}$ tracer's transneuronal transmission. The identical viral particle structures of the H129$_{Amp}$ tracer and helper enable them to co-infect the same neuron, and their shared replication properties results in their synchronized genome replication and protein synthesis following an intracranial single injection. This simplifies experimental operation and reduces potential experimental failures that would occur following lack of precise spatial overlap of helper and tracer injections as can occur by sequential injections required with conventional monosynaptic tracer systems.

### Fast tracing

Most current monosynaptic tracers, both retrograde and anterograde, use AAVs as helpers. Due to the slow onset of expression for AAV vectors, the use of AAV necessitates two separate injections, first helper and then tracer, to allow sufficient expression and accumulation to support replication and transmission of the tracer. To complementarily express and accumulate a sufficient amount of the required target viral protein, AAV helper has to be injected 2–3 weeks prior to the tracer's administration. Then, an additional 5–10 days are required to allow the tracer to replicate and transmit. Taking all of these steps together, conventional monosynaptic tracing usually takes 3–4 weeks. In contrast, the H129$_{Amp}$ tracer system achieves anterograde monosynaptic tracing and postsynaptic neuron labeling in as short as 5 days by simultaneous administration and mutual replication. This significant design improvement will accelerate neuroscience research and will streamline neural connectome mapping.

### Bright labeling

The H129$_{Amp}$ pseudo-genome is of similar size compared to the wild-type H129 genome (~152 kb) and contains multiunit expression cassettes in a concatemer form[34]. When the H129$_{Amp}$ tracer infects neurons and releases the pseudo-genome, multiunit expression cassettes transcribe the target genes simultaneously. This leads to rapid accumulation of the target proteins, including the fluorescent protein and directly contributes to strong labeling signal in postsynaptic neurons.

### Large packaging capacity

Most gene delivery vectors and neural circuit tracers have very limited genetic payloads (~4.8 kb for AAV, ~8 kb for lentivirus and ~4 kb for G deleted rabies virus (RVdG))[4]. YFV-17D has a single-stranded positive-sense RNA genome of ~11 kb[35], which means that it also has a limited vector capacity. However, the pseudo-genome of H129$_{Amp}$ tracer has a much larger genetic payload capacity of 150 kb, thus enabling delivery of either single large genes or multi-gene clusters[20]. This payload capacity feature of the HSV amplicon has attracted great interest from many researchers as a promising gene delivery vector. Using the amplicon vector, a multi-gene cluster of 25.6 kb was delivered into the striatum of a rat model of Parkinson's disease with high-level expression for long-term biochemical and behavioral improvement of Parkinsonian symptoms[36]. Similarly, the H129$_{Amp}$ tracer created in the present study can also be applied to deliver large genes or other functional elements in addition to marker fluorescent proteins.

### Low toxicity and improved potential for functional mapping

Conventional H129-derived tracers display strong toxicity in infected neurons and so they cannot be used to perform functional mapping[9,15]. In the H129$_{Amp}$ tracer system, the H129$_{Amp}$ tracer neither replicates nor expresses any toxic viral protein after transmitting to target postsynaptic neurons, thus displaying minimized toxicity. This unique feature of the H129$_{Amp}$ tracer system enables functional assays on the 2nd-order neurons, such as electrophysiological experiments, opto- or chemo-genetic assays, calcium imaging, etc. The potential for functional mapping is another significant improvement for anterograde monosynaptic tracer applications.

### Tracing postsynaptic neurons of input-defined subpopulations

Input-defined postsynaptic neurons' anterograde monosynaptic tracing, from A to B and then to C (3rd order), is required to better understand the neuronal connectome anatomically and functionally. This has been termed as TRIO (trace the relationship between input and output) or cTRIO (conditional TRIO)[37]. TRIO has been performed by the rabies virus tracer (RVdG and AAV-G together with canine adenovirus or retrograde AAV) and transsynaptic AAV2/1 (together with another AAV reporter)[10,38–40]. However, these tracers transmit through a single synapse instead of two sequential synapses. In the present study, input-defined postsynaptic neurons' anterograde monosynaptic tracing by using the H129$_{Amp}$ tracer system is achieved in combination with another tracer set (H129-dgK-G4 and AAV2/9-DIO-mCh-gK). We are continuing our work on developing the Cre-specific input-defined postsynaptic neurons' anterograde monosynaptic tracer systems and expect we can make further refinements and performance improvements. Input-defined postsynaptic neurons' monosynaptic anterograde tracing from the Cre$^+$-neuron could be achieved by combining the H129$_{Amp}$-Flp-DIO-TG tracer system and Flp-dependent H129-dgK tracer set (AAV2/9-fDIO-mCh-gK and H129-dgK-G4)

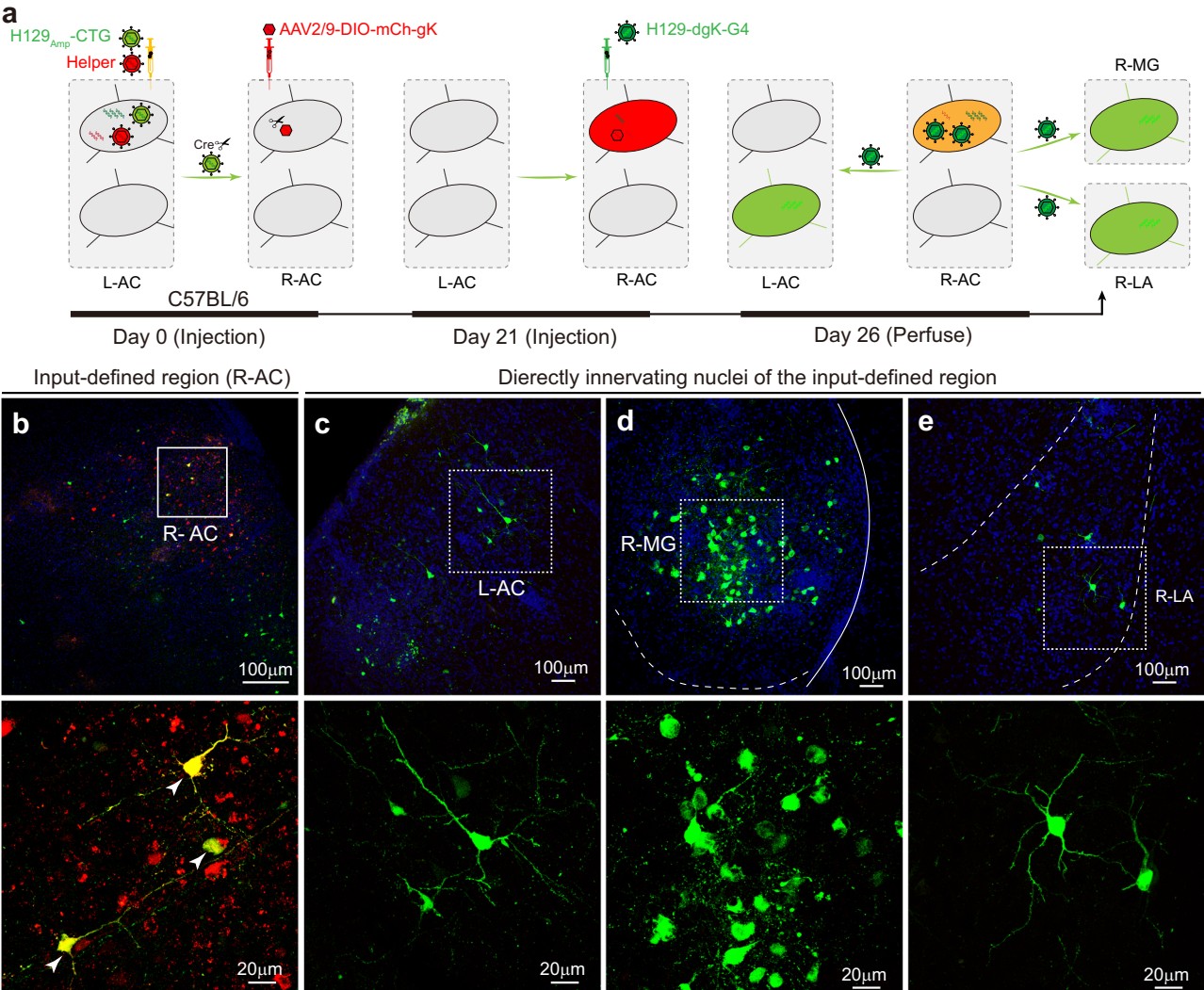

**Fig. 4 | Input-defined postsynaptic neurons' anterograde monosynaptic tracing with H129$_{Amp}$ tracer system together with other tracers. a** Schematic illustration of the experiment setup and tracing strategy. The H129$_{Amp}$ tracer system (H129$_{Amp}$-CTG $1.5 \times 10^8$ pfu/ml and helper $1.5 \times 10^8$ pfu/ml, in 300 nl) was injected into the L-AC (AP: −2.80 mm; ML: −4.13 mm; DV: −2.38 mm) of wild-type C57BL/6 mice, and AAV2/9-DIO-mCh-gK ($1.0 \times 10^{12}$ vg/ml, 100 nl) was simultaneously injected into the R-AC (AP: −2.80 mm; ML: +4.13 mm; DV: −2.38 mm). At Day 21, H129-dgK-G4 ($5.0 \times 10^8$ pfu/ml, 100 nl) was injected into the R-AC of the same mice. Brains were collected at Day 26, and images were obtained after cryosection and DAPI counterstaining. H129$_{Amp}$-CTG propagates in the L-AC neurons (1st order) with helper assistance, and transmits through the first synapse to the 2nd-order neuron in the R-AC. There, H129$_{Amp}$-CTG expresses Cre to initiate AAV2/9-DIO-mCh-gK expressing mCherry and gK, labeling the neurons and supporting H129-dgK-G4 propagation, respectively. The newly produced H129-dgK-G4 then transmits through the second synapse to the 3rd-order neurons, labeling them with GFP. **b−e** Representative input-defined postsynaptic neurons' anterograde monosynaptic tracing result at Day 26. Shown are the representative images of the 2nd-order brain region, R-AC (**b**), and the 3rd-order regions, including L-AC, R-MG, and R-LA (**c−e**). The 1st-order starter neurons were not visible anymore and the potential 2nd−order starter neurons are labeled by both GFP and mCherry (merged as yellow), indicated with white arrowheads (**b**). Images with higher magnifications of the boxed areas are presented in the lower panels.

(Supplementary Fig. 10a). To trace the outputs from the input-defined Cre$^+$-neuron, Flp-expressing tracer (H129$_{Amp}$-FTG) and FRT-floxed-*pac* modified helper (H129-dTK-T2-FRT-pac-FRT), together with Cre/ Flp-double-dependent H129-dgK tracer set (AAV2/9-$_{Con}$F$_{on}$-mCh-gK and H129-dgK-G4) are required (Supplementary Fig. 10b, c). We acknowledge that given the efficiency requirements for two successful transsynaptic transmissions, this strategy might not work when Cre$^+$ neurons are of very low abundance.

## Limitations
Despite the above-described outstanding advantages, our newly developed H129$_{Amp}$ tracer system still has a few remaining limitations, including the limited observation time window and toxicity in the starter neurons. HSV amplicon expresses the genetic payload rapidly and massively, but the transcription is silenced quickly[41,42]. Infection of

HSV amplicon may initiate a cascade of immune responses including expression of Type I IFNs, which have been reported to suppress transgene expression at the transcriptional level by activation of STAT1[41,42]. In addition, the presence of prokaryotic DNA sequences in the pseudo-genome of H129$_{Amp}$ tracer may also lead to transcriptional silencing of the entire vector sequence due to the inactive form of chromatin immediately after infection[19,29]. The transgene expression kinetics of HSV amplicon has been shown to peak quickly (1 day post-injection, dpi) and then drop by over 3 logs within a week[19]. We observe fluorescence labeling of the H129$_{Amp}$ tracer system as early as 1 day post-injection at the injection site, but this signal quickly dims below the detection threshold for confocal imaging at 7 days post-injection. Therefore, the time window for observing tracing results is limited between Day 3 to Day 5 following injection. A better reporter may extend the observation time window to 21 days or even longer, such as

Ai14 mice or reporter AAV (AAV2/9-DIO-ChR2-mCh and AAV2/9-fDIO-ChR2-mCh). Other improvements could be made by adding insulator and AT-rich sequences or by removing the bacterial sequence in the amplicon plasmid. These modifications could extend the labeling duration[19]. Second, toxicity does occur in the starter neurons that are co-infected by H129$_{Amp}$ tracer and helper. The tracer and helper act cooperatively to support each other replication and this leads to high levels of viral protein expression and strong toxicity in the starter neurons. This presents a remaining key obstacle that prevents functional assays on starter neurons. Third, the system theoretically may lead to anterograde multi-synaptic transmission. Hypothetically, a small fraction of helper, whose floxed-*pac* is not excised by Cre, could be packaged and transmitted to the postsynaptic neurons with the tracer. In the 2nd-order neurons, such helper could assist the tracer to transmit to the further downstream neurons. However, we see no evidence for helper transmission to postsynaptic neurons or evidence for tracer transmission to further downstream neurons even in the highly sensitive Ai14 reporter mice. We encourage users to apply proper controls such as tracer/helper titration to exclude the spurious actifactual labeling that might be caused by unintended anterograde multi-synaptic transmission. Further, it is theoretically possible that tracer could be transmitted from a local Cre⁻ neuron which is infected by helper and recombined tracer (losing Cre-dependency) and transmitted from a nearby Cre⁺ neuron when H129$_{Amp}$-DIO-TG tracer system is applied for starter-specific tracing. To determine whether nonspecific transmission occurs, we tested for this possibility in PV-Cre mice that express Cre-recombinase in parvalbumin (PV) interneurons. The H129$_{Amp}$-DIO-TG tracer system was injected into the AC of PV-Cre mice with optimized tracing titer, and the results were examined at Day 5 (Supplementary Fig. 11a). PV neurons (Cre+) in AC make only local cortical connections, but some other Cre⁻ neurons nearby may hypothetically form long-range projection (Supplementary Fig. 11b). However, GFP (green) labeled neurons are only observed at the injection site, but not detected in any other examined brain regions (Supplementary Fig. 11c–h). These results indicate that Cre⁻ neuron-initiated transmission does not occur in our study. While we cannot exclude the possibility of this happening, we recommend this should be taken into account by other investigators for tracing applications and accounted for by setting up careful controls that include careful calibration of titer. Currently, we are working on solving this potential issue by modifying the helper to express gK in a Cre-dependent manner (H129-dTK-DIO-gK-T2-*pac*$^{Flox}$).

In summary, we have developed innovative strategies to create an H129$_{Amp}$-based anterograde monosynaptic tracer system for connectome mapping. The unique advantages of our H129$_{Amp}$ anterograde monosynaptic tracer system include simplified operation, fast tracing, bright labeling, large capacity, low toxicity in target neurons, improved potential for functional mapping of target neurons, and input-defined postsynaptic neurons' anterograde monosynaptic tracing. This H129$_{Amp}$ tracer system will greatly benefit further neuroscience research.

## Methods
### Animals and ethics
Wild-type C57BL/6NCr1 mice (C57BL/6) were purchased from Vital River Laboratory Animal Technology Company (Beijing, China). Ai14 reporter mice (JAX-007908), Rph3a-Cre mice (with P2A-iCre behind the *Rph3a* coding gene), PV-Cre mice (JAX-017320), and CaMK2a-Cre mice (JAX-005359) were kindly provided by the Laboratory Animal Resource Center at the Chinese Institute for Brain Research (Beijing, China), the School of Basic Medicine, Huazhong University of Science and Technology (Wuhan, China), and the Eye Center of Renmin Hospital, Wuhan University (Wuhan, China). All mice received food and water ad libitum in a 12 h light/dark cycles and were maintained under conditionals of stable temperature (23–25 °C) and consistent humidity

(45–55%). All mice used were the C57BL/6 background strains of either sex at 8–12 weeks old. All experiments with mice were reviewed and approved by the Institutional Review Board and Institutional Animal Welfare Committee of Wuhan Institute of Virology, Chinese Academy of Sciences (WIVA10201502). All experiments with viruses were reviewed and approved by the Institutional Biosafety Committee, and performed in Biosafety Level 2 (BSL-2) laboratory or Animal Biosafety Level 2 (ABSL-2) facilities following the Institutionally approved standard operating procedures and biosafety guidelines.

### Cells and cell culture
Vero-E6 cell (Vero, ATCC# CRL-1586) and 293 T/17 cells (293 T, ATCC# CRL-11268) were purchased from ATCC, maintained in the laboratory, and tested to be mycoplasma-free. Cells were cultured with DMEM containing 10% fetal bovine serum and penicillin-streptomycin (100 U/ ml and 100 µg/ml) (Gibco/Life Technologies, USA) in a humidified incubator with 5% CO$_2$ at 37 °C.

### Construction of the H129$_{Amp}$ tracer system
*Ori* (the viral genome replication origin) and *pac* (the genome packaging signal) were amplified from the H129 bacterial artificial chromosome (H129-BAC) by PCR[7], cloned into pcDNA3.0, and generated the H129 amplicon backbone plasmid pHSV. Then, different expression cassettes, including the recombinases (Cre or Flp), HSV tyrosine kinase (TK) gene, and GFP, were further inserted into pHSV, resulting in a serial of H129 amplicon plasmids. The structure of the amplicon plasmid pHSV-CTG was shown as representative (Fig. 1a).

The helper, H129-dTK-T2-*pac*$^{Flox}$, was derived from H129-dTK-T2, a TK deficient recombinant virus containing 2-copy tdTomato expression cassette[14]. Based on the previously introduced H129-dTK-T2-BAC[43], one *pac* was deleted (Δ*pac*) and the other *pac* was flanked by loxN (LoxN-*pac*-LoxN, *pac*$^{Flox}$) via *galK*-based homologous recombination (Fig. 1b). Finally, the resulted recombinant virus H129-dTK-T2-*pac*$^{Flox}$ was reconstituted and propagated in Vero cells following the protocol described previously[14], and applied as the helper for further H129 amplicon (H129$_{Amp}$) production and monosynaptic tracing.

To produce the H129$_{Amp}$ tracer system, represented by H129$_{Amp}$-CTG, $2 \times 10^5$ Vero cells in a 100 mm tissue culture dish were transfected with 20 µg of pHSV-CTG. After 24 h, H129-dTK-T2-*pac*$^{Flox}$ was inoculated to the transfected cells at a multiplicity of infection (MOI) of 1 to rescue the H129$_{Amp}$-CTG tracers (Fig. 1c). The supernatant, containing the H129$_{Amp}$-CTG tracer and H129-dTK-T2-*pac*$^{Flox}$ helper, was collected at 48 h post-infection (hpi) and served as the seed for further propagation. To propagate high-titer H129$_{Amp}$ tracer, the seed was mixed with H129-dTK-T2-*pac*$^{Flox}$ and then inoculated to Vero cells to reach final MOIs of 0.1 (for H129$_{Amp}$-CTG) and 1 (for H129-dTK-T2-*pac*$^{Flox}$), respectively. The supernatant was harvested at 48 hpi and concentrated as described previously[14]. Limited by the recombination efficiency of Cre-recombinase, a very small fraction of the replicated helper genome, whose floxed-*pac* is not excised, packaged and forms the helper "contamination" (~5%) in the raw tracer product. Then, the titers of the H129$_{Amp}$-CTG tracer and the H129-dTK-T2-*pac*$^{Flox}$ helper in the raw tracer product were determined by plaque-forming assay based on GFP and tdTomato, respectively.

Finally, the independently propagated H129-dTK-T2-*pac*$^{Flox}$ helper was added to the titrated raw tracer product to reach the optimized final working titer as specified. This titer-adjusted tracer/helper mixture, designated as the H129$_{Amp}$ tracer system, was aliquoted and stored at −80 °C until administration.

### Other applied viral tracers
The recombinant AAVs applied in the present study were created and packaged in 293 T cells following the standard protocol, and stored at −80 °C as aliquots until application[44]. AAV titers were determined according to the copy number of viral genome (vg) by absolute

quantitative PCR[44]. The monosynaptic tracer H129-dgK-G4 was propagated in the Vero-gKmut cell line and was applied for tracing[9]. After titration using a plaque-forming assay, H129-dgK-G4 was adjusted to $5.0 \times 10^8$ pfu/ml, aliquoted, and stored at $-80\,^{\circ}\text{C}$ until use.

### Intracranial injection and imaging
The H129$_{Amp}$ tracer system and other viral tracers were intracranially administrated into the target brain regions by stereotaxic injection[14]. The exact coordinates of the injected sites were determined according to the Mouse Brain Atlas (second edition) by the mediolateral (ML), anteroposterior (AP) and dorsoventral (DV) distances to Bregma[45]. The brains were collected after perfusion with 4% paraformaldehyde at the indicated times. Samples were processed and coronally cryo-sectioned to 40 μm thick slices using a cryostat Microme (HM550, Thermo/Life Technologies, USA). The brain slices were counterstained with DAPI (Roche, Switzerland), and then imaged using a Nikon's A1R MP+ confocal microscope equipped with a fast high-resolution galvanometer scanner[14].

### In vitro optogenetics and electrophysiological assays
The subjected mouse brains were carefully collected under anesthetization with overdosed pentobarbital (0.7%, 0.1 mL/g body weight), and immediately sectioned to 300 μm thick coronal slices using a vibrating microtome (Leica VT1200s)[46]. The slices were recovered by incubation in a submersion chamber filled with the prewarmed (35 °C) artificial cerebrospinal fluid (126 mM NaCl, 2.5 mM KCl, 1.25 mM NaH$_2$PO$_4$, 26 mM NaHCO$_3$, 10 mM D-Glucose, 2 mM MgSO$_4$, 2 mM CaCl$_2$) for 30 min, and then gradually cooled to the room temperature. The electrophysiological assay was performed by patch-clamp recordings from AC neurons in slices visualized under a fluorescent infrared-phase-contrast (IR-DIC) Axioskop 2FS upright microscope. The internal solution in the recording electrodes consisted of 140 mM potassium gluconate, 10 mM HEPES, 0.2 mM EGTA, 2 mM NaCl, 2 mM MgATP, and 0.3 mM NaGTP[47]. Electrophysiological recordings were obtained from Au neurons with Multiclamp 700B amplifiers and pCLAMP 10.3 software. To record light-evoked excitatory postsynaptic currents, 5 ms pulses of 5 mW blue light (DPSS laser, Anilab) were delivered through the objective to ChR2-expressing axons originating from the indicated brain region(s).

### Statistics and reproducibility
Data were collected from at least three independent experiments or animals, each experiment was performed in triplicate, and the results were presented as means ± SEM (Standard Error of the Mean) using GraphPad Prism 9.

### Reporting summary
Further information on research design is available in the Nature Portfolio Reporting Summary linked to this article.

## Data availability
The plasmids, tracers, and helpers are available from the corresponding authors upon request. The raw imaging data are available upon request from the corresponding authors. Source data are provided with this paper.

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

## Acknowledgements

We thank Hai-Wen Zhao for the technical support on electro-physiological assays. This work was supported by the National Natural Science Foundation of China (81620108021to H.M.L., 82172264 to H.M.L., 32070169 to B.W.Z., 81571355 to F.Z., and 81601206 to B.W.Z.).

## Author contributions

F.X., F.Z., X.X., T.H., W.B.Z., and M.H.L. wrote the manuscript with data contributions from all co-authors. W.B.Z. generated the key components of H129-dTK-T2-*pac*^Flox and pHSV-CTG. F.X. and Y.H. performed the experiments, collected, and analyzed the data. H.B.Q. and Q.Y.Z. helped with the experiments and data collection. Y.G.S. established the methods of HSV-1 amplicon production. W.J., M.D., Y.L. (Yang Liu), X.H., and Z.L. designed and performed the optogenetic and electrophysiological experiments, with Y.S., Y.H., Y.L. (Youming Lu), and M.L.'s instruction. M.H.L., W.B.Z., and F.Z. supervised the project. All authors read and approved the final manuscript.

## Competing interests

The authors declare no competing interests.
