## [Peer Review File · Nature Communications]

An HSV-1-H129 amplicon tracer system for rapid and efficient monosynaptic anterograde neural circuit tracingReviewers' Comments:

Reviewer #1:

Remarks to the Author:

Feng Xiong et al present a new experimental approach for anterograde neurotracing with HSV129 vectors that reduces the experimental timeline from several weeks to under one week. The authors claim that their amplicon helper virus system overcomes several of the problems associated with existing approaches of anterograde labeling that utilize HSV namely low labeling intensity and high toxicity in postsynaptic neurons. In general the data do support this claim and demonstrate the potential of the amplicon system. The idea behind this new approach is pretty clever and the amplicon system could have a significant impact in the field even though it does not solve the problem of high toxicity in 1st order neurons.

However, it is my understanding that 1st order neurons should be either green or double -labeled as presented in figure 1c. In practice the authors observe numerous 1st order neurons that are labeled brightly red (Fig.2c). This suggests that the system does not always work as it is presented in Figure 1c. The field has seen other ingenious approaches published in high impact journals that looked quite promising but have not found widespread use presumably because of experimental difficulties and problems with reproducibility. To allow a better assessment of the amplicon system in particular with regards to its reproducibility the authors should present a summary of the number and percentage of double labeled cells from several experiments like the one shown in Fig. 2c. I would also like to see some data on the robustness of this system that is how sensitive it is to changes in the Amplicon/Helper ratio.

Reviewer #2:

Remarks to the Author:

This manuscript describes the development and testing of several new reagents for anterograde trans-neuronal tracing. While the results presented clearly show that the new reagents will have utility for experiments that are not possible with other existing tracers, there are many concerns about various claims that are made without experimental support, and/or potential limitations or labeling artifacts that the authors do not consider. The manuscript would also benefit considerably from clearer descriptions of the various reagents and how they are expected to behave and interact with each other. (The Overview below provides an example of how to do this.) This review begins by providing an overview of the various reagents used and their apparent intended uses. Then the review turns to describing possible limitations to the methods that the authors either fail to consider or claim to have considered but do not support with definitive evidence. The review then turns to problems with organization of the manuscript or lack of detail in descriptions that make it difficult to follow and which could be remedied by some re-writing. Various edits are suggested in the context of the original text. Finally, there are several minor points that could be addressed.

Overview of the reagents described and their intended uses:

The manuscript describes three different tracing systems. All three have in common that they utilize an HSV-H129 strain helper virus to allow the production/packaging of an amplicon vector ("tracer") that then spreads anterogradely to postsynaptic neurons. The helper is called H129-dTK-T2-LoxN-pac-LoxN. dTK refers to deletion of TK from the genome. TK is required for viral replication in dividing cells, such as neurons. T2 refers to the insertion of genes for TdTomato such that infected cells will express TdTomato. LoxN-pac-LoxN refers to the fact that the pac sequence is floxed allowing for its removal following Cre-recombination. This is relevant because pac is required for packaging of viral genomes into viral particles. There are two different tracers used in these experiments as well as two different H129 viruses. How each is used depends on experimental goals of the 3 different systems.

The first system is intended to identify the directly postsynaptic neurons downstream of neurons infected in a brain area of interest. It does not distinguish between outputs of different neuron types in the brain area of interest. Here the helper is H129-dTK-T2-LoxN-pac-LoxN and the tracer is H129ampCTG, which expresses Cre-recombinase, TK, and GFP. These are mixed together and

co-injected into the region of interest. In neurons that are co-infected, the helper genome is recombined by Cre in order to remove pac and prevent trans-neuronal spread of the helper. Expression of TK allows the helper to generate all of the required viral structural components and package the tracer genome into amplicons which are then transmitted down the axons and trans-neuronally to downstream recipients. GFP expression from the tracer then marks the downstream neurons. Insofar as pac is removed from the helper genome, the helper will not spread beyond the initially infected neurons. Furthermore, the tracer is unable to spread beyond the first synaptic step because it is unable to replicate in the absence of helper. The absence of helper from the downstream neurons is also expected to mitigate toxicity. GFP expression from the amplicon tracer is expected to be transient (cite references). If this approach is employed in a cre-reporter mouse line, such as AI14, expression of Cre from the amplicon tracer can permanently label downstream neurons.

The second system is intended to identify the directly postsynaptic neurons downstream of a selected population of neurons that express cre-recombinase (from a mouse driver line) in a brain area of interest. It is intended that the outputs of neighboring cre-negative neurons in the region of interest will not have their outputs labeled (but see below). For these experiments, the same helper is used (H129-dTK-T2-LoxN-pac-LoxN) but the amplicon (H129amp-DIO-TG) expresses only TK and GFP (not Cre), and their expression requires Cre-recombination. When this helper and tracer are co-injected in an area of interest of a cre driver line, pac will be removed from the helper genome in Cre+ cells, preventing it from spreading. Cre will also recombine the amplicon, allowing it to express TK such that the helper can replicate and facilitate production of tracer amplicons that spread to and express GFP in downstream neurons. Cre-negative neurons infected with helper and tracer will initially lack TK and will therefore not be able to produce amplicon tracer. Therefore, the downstream neurons of Cre- neurons would not be labeled. Note however, that for brain regions in which Cre-positive and Cre-negative neurons are interconnected (such as auditory cortex), labeling of downstream neurons will not actually be selective for those connected to the Cre-expressing neurons, as intended. This is because TK expressing amplicons will spread to Cre-negative neurons allowing the helper to be active and produce tracer amplicons in the Cre-negative neurons. See further below.

The third system is intended to trace circuits anterogradely across two synaptic steps. It uses the first system described above (H129-dTK-T2-LoxN-pac-LoxN helper and H129ampCTG tracer) to initiate transfer of the tracer to downstream targets. In addition, a second H129-based viral tracer that was described in a previous publication (H129-dgK-G4) is injected into a downstream region. H129-dgK-G4 has an envelope protein required for packaging infectious particles deleted and expresses GFP. It can be transcomplemented by expression of gK to mediate monosynaptic anterograde spread. In order to trace connections across a second synaptic step an AAV helper virus with Cre-dependent gK is injected into the downstream region of interest at the same time that the H129-dTK-T2-LoxN-pac-LoxN helper and H129ampCTG tracer are injected. The H129-dgK-G4 is injected 3 weeks later. In the downstream target region H129ampCTG tracer that has spread to postsynaptic neurons expresses Cre and recombines the AAV helper. The resulting gK expression allows the H129-dgK-G4 to spread an additional synaptic step resulting in GFP label.

Possible limitations and labeling artifacts that are not considered or for which claims against them are not supported:

The description above describes how it is hoped that the tracing systems will work. But there are many possible scenarios not considered by the authors or not ruled out by results presented, that would result in very different undesired outcomes. Below, these are each considered in turn with a description of the relevant reagents and experiments and data presented that are relevant.

- 1) For tracing system 1, it is claimed that helper virus does not spread from the initially infected neurons and that, as a consequence, the spread of tracer is monosynaptic (not multi-synaptic spread). The validity of this claim relies on the unproven assumption that Cre-recombination from the tracer will successfully recombine and remove pac from every copy of the helper genome. This seems unlikely given that the helper will rapidly replicate in the presence of TK expressed from the tracer. To support these claims it is necessary to show that there is no TdTomato expressed at postsynaptic target regions at very short times (1 or 2 days) after injection of the tracer system.

For example, following injections of the tracing system into AC, please show photographs of the MG at d1 and d2 post-injection. Are there any red cells observed? Photographs at d5 are not adequate because it is clear that all cells infected with helper would have been killed and eliminated at d5. This is clearly shown in Fig. 2d which illustrates that red fluorescent cells have disappeared from the AC between d1 and d5. (Compare to Fig. 1c.)

If there are red cells in postsynaptic targets of the AC at d1, then it is clear that they could mediate spread of tracer another synaptic step. Please show results that clearly illustrate a lack of spread in cases where it might be expected. For example, the results state: "... but not in further downstream second order connected nuclei (data not shown). These results show that H129Amp tracer anterogradely labels directly connected postsynaptic neurons limited to monosynaptic connections." It is not possible to show something with results that are not shown. Please specifically state what structures were observed that might have potentially been labeled by multisynaptic spread. Such structures should contain the GFP-labeled axon terminals of neurons that were trans-neuronally labeled with the tracer. But they should not contain labeled postsynaptic neurons. Please provide photographs illustrating the presence of axon terminals in these locations without postsynaptic labeling.

2) The authors do not consider the known fact that both the helper and the tracer are packaged with envelope proteins that will mediate retrograde infection. The H-129 strain is biased toward anterograde because it is trafficked down the axons of infected neurons. But there is not any mechanism that prevents infection of axon terminals and retrograde spread to the infected cell bodies. As a result, it is expected that injections of the tracing system into cortical locations (such as AC) will result in direct infection of thalamic neurons (such as in MG). This should be clearly visible as TdTomato labeling in the MG following injections of either tracer or helper alone (or both together) into AC. Please show evidence that none of these occurs. For example, show photographs of the MG at day 1 after injections of the tracing system into AC. Are there any red neurons in the MG. Again, note that the absence of red label in MG at day 5 is not sufficient because directly infected neurons are clearly dead and disappear by d5. Please also show evidence that tracer does not retrogradely infect and express GFP in the absence of helper. This is a crucial control. But surprisingly there are no data presented in the manuscript showing the nature of direct infection with the tracer in the absence of helper. Please show photos of the AC and other pre-and postsynaptic targets of AC (e.g. SC, MG, contralateral AC) following injections of tracer without helper.

3) The second "starter-specific" tracing system is unlikely to actually be specific as intended and claimed. This system utilizes a Cre-dependent tracer (H129Amp- DIO-TG) and is intended to trace the monosynaptic outputs selectively from Cre-expressing cells. The authors do not describe the rationale behind how this should work, but apparently they expect that there will not be any expression of TK in cells that do not express Cre. And since Cre-negative cells at the primary injection site are infected with helper but do not express TK, the helper will not replicate in Cre-negative cells. It is also assumed that tracer will not spread locally between Cre-positive and Cre-negative cells. But these assumptions are certainly wrong for the cortical experiment shown. This is because there are direct connections between the Cre-positive and the Cre-negative cortical cells. It is therefore unlikely that the postsynaptic label observed in distant structures reflects selective outputs from the Cre-positive cortical cells. The expected mechanism for non-selective spread is as follows. Following injections of the tracer and helper into the cortex, both vectors infect both Cre+ and Cre- neurons. In the Cre+ neurons, pac is removed from the helper and the tracer expresses TK and GFP. pac is not removed from the helper that is in the Cre- neurons. Tracer spreads from the Cre+ to connected Cre- neurons where it expresses TK and allows replication of the helper and further production of tracer amplicons in the Cre-negative neurons. This allows the tracer to spread an additional synaptic step and to label the outputs of Cre-neurons. In addition, helper with intact pac is also replicated and can spread from the Cre-neurons, resulting in further multisynaptic spread.

The authors do not appear to have considered this and did not use a Cre-expressing mouse line that is capable of demonstrating this artifact. For example, it is expected that if the experiment were conducted in the AC of PV-Cre mice expressing Cre in parvalbumin inhibitory neurons (which make only local cortical connections) "starter-specific" tracing would not result in spread outside

the cortex. But the artifact described would allow the spread of tracer from PV interneurons to cortical excitatory neurons, and then from the excitatory cells to postsynaptic neurons in more distant targets. It is necessary to show results from a Cre line in which it is possible for this artifact to be revealed and evaluated. Another such Cre line would be Tlx3-Cre which expresses Cre in cortical layer 5 pyramidal neurons that make cortico-cortical projections but not those that project to subcortical structures. If the spread is actually "starter-specific", use of the system in this line should result in anterograde tracing to cortical targets but no labeling in subcortical structures such as superior colliculus.

Other comments/suggestions:

"Although multiple anterograde monosynaptic tracers have been developed based on the herpes simplex virus 1 (HSV-1) strain H129 (H129), functional mapping for output connectome requires further technical development." Suggest changing to "...connectome would benefit from further ...".

"Both retrograde and anterograde monosynaptic tracing are achieved by combining two components: i) a viral tracer (tracer), which is a replication- or transmission-deficient virus with one or more genes deleted whose trans-neuronally infects connected neurons requires the assistance of ii) a helper virus (helper) which complementarily expresses the tracer's deficient gene(s), thus supporting tracer replication and transmission." Suggest the following edits: "Both retrograde and anterograde monosynaptic tracing are achieved by combining two components: i) a conditionally competent viral tracer (tracer) and ii) a helper virus (helper). The viral tracer is replication- and/or transmission deficient due to the deletion of one or more genes that are required for transneuronal spread to connected neurons. The helper virus expresses genes which replace or complement tracer's deficient gene(s), thus supporting tracer replication and/or transmission."

The following sentence is factually incorrect and needs to be changed. "The AAV helper has limited expression efficiency due to a single copy of target gene and deficiency in replication, thus needs to be administrated first of two separate injections to allow sufficient target expression and accumulation to support replication and transmission of tracer." AAV genomes concatenate to incorporate many gene copies into episomes. Thus, AAV is not limited to a single copy, and is capable of expressing at very high levels. It is more relevant to this manuscript that expression is slow and that AAV has a limited payload. Suggest changing to: "Due to the slow onset of expression for AAV vectors, the use of AAV necessitates two separate injections, first helper and then tracer, to allow sufficient expression and accumulation to support replication and transmission of the tracer."

The introduction provides no context for the statement: "Moreover, for the first time, input-defined postsynaptic neurons' monosynaptic anterograde tracing is succeeded by combining H129Amp tracer system with H129-dgK-G4 tracer." For the reader to understand this statement it will be necessary to describe the H129-dgK-G4 tracer. What are the expected advantages (if any) of using H129-dgK-G4 tracer with H129amp tracer? How is it possible to combine these for cell type specific output labeling since H129-dgK-G4 tracer expresses pac and is not disabled by cre-dependent pac removal?

There is nothing in the Introduction or the Results description of generation, production and tracing principles to make it clear that, for the first tracing system described, the initial infection or production of tracer is not cell type specific. Please state clearly that the initial injection of tracer and helper will infect and spread from all types of neurons at the injection site.

"Abundant AC neurons were labeled by H129Amp tracer (green) and/or helper (red)." Why are neurons labeled red? Is there a red fluorescent protein expressed from the helper genome? Is the tissue stained with an antibody against the helper? There is nothing in the introduction or the results description of generation, production and tracing principles to indicate that TdTomato is expressed from the helper genome. This can only be guessed by looking at Figure 1b. There is also nothing in the figure 1 legend to indicate that this guess is correct. The following edits are suggested. At line 125 rewrite as: "The H129Amp tracer system is composed of H129Amp tracer

and H129-dTK-T2-LoxN-pac-LoxN helper. H129-dTK-T2-LoxN-pac-LoxN helper has the thymidine kinase gene deleted (dTK), expresses two copies of the fluorescent protein tdTomato (T2), and its pac gene can be deleted by Cre-recombination (LoxN-pac-LoxN). Due to the identical infection features of H129amp tracer and helper, the system it requires only a single-injection instead of the conventional sequential twice-injection." Also update the figure 1 legend to clearly indicate what is schematized in 1b.

The ED Fig. 2 legend and the figure refer to LP. Based on the photograph it appears that the location indicated by LP is the lateral pulvinar nucleus of the thalamus, which receives direct input from SC. But the figure legend states: "LP, pretectal area;". This is probably wrong.

"These results show that the postsynaptic neurons labeled by H129Amp tracer system maintain normal physiological conditions, thus allowing for functional connection mapping." This result cannot be generalized to all neurons labeled with the tracer. This only applies to cells that are still present after 14 days. Other cells may have died.

"Notably, H129Amp tracer system is also capable of achieving input-defined outputs mapping from specific starter neurons, which cannot be done using transsynaptic AAV2/1." There is no reason to expect that this could not be done with transsynaptic AAV2/1. If AAV2/1 expressing Cre is used in a mouse with Cre-conditional Chr2 expression (either from the genome or from an AAV-DIO-Chr2 injected at the postsynaptic site) then the same experiment could be done.

"Fast tracing. All current monosynaptic tracers, both retrograde and anterograde, use AAVs as helpers." This statement is not true. For example, monosynaptic rabies tracing has been conducted using mouse lines that have cre-dependent or tTA-driven expression of the helper genes. Thus, there is no need for injection of AAV helper or to wait for expression of helper genes.

"In addition to the single-copy target, deficiencies in the replication of viral genome and production of viral particle further limit the expression efficiency of AAV genetic payload." As noted previously, AAV is able to express its payload very efficiently and without toxicity. Replication is not necessary for efficient long-term expression from AAV vectors.

"In the novel H129Amp tracer system, helper, the toxic viral protein producer, is left behind in the starter neurons." As noted above, there are no data presented to support this assertion. It is plausible that the Cre expression from the tracer does not completely remove cap from all helper-infected neurons and that some helper spreads trans-neuronally, along with the tracer. Subsequently, the co-infected trans-neuronally labeled cells die. This is consistent with the observation that GFP expression does not persist in labeled postsynaptic neurons. Please explain why GFP expression is not persistent. I was able to find this in papers cited but readers should not have to track down the relevant facts.

Recent improvements in HSV amplicon vectors might allow more stable gene expression. For example see Soukupova et al., 2021; Improvement of HSV-1 based amplicon vectors for a safe and long-lasting gene therapy in non-replicating cells. [https://www.cell.com/molecular-therapy-family/methods/fulltext/S2329-0501\(21\)00060-7](https://www.cell.com/molecular-therapy-family/methods/fulltext/S2329-0501(21)00060-7). This might be worth mentioning in the discussion.

Reviewer #3:

Remarks to the Author:

In this study, Xiong et al. developed a novel HSV-1-H129 amplicon tracer system as a monosynaptic viral tracer for dissecting neuronal connectomes and targeted delivery of molecular sensors and effectors. The improvements of the H129Amp tracer system shorten the experiment duration from 28-days to 5-days for fast-monosynaptic tracing and minimize toxicity in the postsynaptic neurons. The idea behind this study is very exciting. However, some of the experiment's design and interpretations need more data and or to be further clarified and developed.

1. The authors stated:" For anterograde monosynaptic tracing, the tracer system was

administrated into the brain region of interest by a single-injection. In neurons, H129Amp tracer does not replicate alone since it contains no viral gene, but it efficiently expresses Cre, TK, and GFP with its multiunit cassettes in the concatemeric pseudo-genome (Fig. 1d)." It is possible that the GFP can be present in other brain areas to a retrograde transfection mechanism and or transneuronally.

2. The authors should provide data images that include a single injection of H129Amp tracer, showing that the GFP expression is only visible in the injection site and not outside the injection site at the least Day7 and Day 21 post-injection.

3. The same experiments need to be repeated for the helper.

4. Figure 2a and b provided by the authors make it impossible to determine that the virus was injected into the primary AC.

5. Figure 2c. What is the fraction of the co-transfected neurons compared to the single transfected and in which layers of the AC are located? The figure legend of this figure is very confusing. For example, it is not clear if the panels h1 and h2 are a high magnification of the pyramidal neurons of the contralateral AC.

6. The panel 2h depicts the contralateral AC with only a few neurons. Can the authors provide the number of the neurons that are GFP positive in the Cont AC? This is extremely important for the experiments performed in the next section (Figure3).

7. There is no information about the amount of the virus injected into the brain and the stereotaxis coordinates.

8. The authors also indicated that they had determined empirically that after 7 days, the number of the potential co-labeled (potential starter) decreased and dimmed. Can the authors further explain this? Are the neurons dead?

9. The authors stated that: " Many GFP+ neurons were clearly observed in downstream nuclei are directly innervated by AC neurons, including contralateral auditory cortex (Cont. AC), lateral amygdaloid nucleus (LA), medial geniculate nucleus (MG), external globus pallidus (GPe) and locus coeruleus (LC) (Fig. 2e-i), but not in further downstream second-order connected nuclei (data not shown). These results show that H129Amp tracer anterogradely labels directly connected postsynaptic neurons limited to monosynaptic connections." These results are not showing that it is due to monosynaptic connections.

10. Can the authors provide more information about the theRph3a-Cre transgenic mice (i.e., cell type, layers, etc.)?

11. The authors stated: " Next, we mapped the outputs of the input-defined neuron subpopulation anatomically and functionally with H129Amp tracer system. H129Amp tracer system (H129Amp-CTG tracer and helper) was administrated into the AC in the left hemisphere (L-AC) of wildtype C57BL/6 mice as described above, and AAV2/9-DIO-ChR2-mCh was simultaneously injected into the right hemisphere AC (R-AC) in the same mice (Fig. 3a). R-AC receives inputs from the L-AC, and projects back to L-AC, as well as the right hemisphere MG and LA (R-MG and R-LA) (Fig. 3a). Newly propagated H129Amp tracer transmits from the L-AC to the postsynaptic neurons in the R-AC and provides Cre allowing the AAV2/9-DIO-ChR2-mCh to express ChR2-mCherry. As we expected, on Day21 the soma of mCherry labeled neurons (mCh+) were clearly observed in the R-AC (Fig. 3b), but not in any other regions (data not shown)." Can the authors provide a better image of the R-AC (figure 3b)? It is impossible from the actual image concluding that it is the R-AC. What is the fraction of neurons that are positive to ChR2? It looks like that there are more ChR2-positive (figure 3b) than GFP-positive (figure 2h) neurons in the contralateral AC. Why?

12. All the Experiment performed in figure 3 and 4 need further experiments before a conclusion like this can be drawn: " Altogether, these results demonstrate that the H129Amp tracer systems (H129Amp-CTG and H129Amp-Flp-DIO-TG tracers) in combination with the appropriate reporter AAVs are capable to not only anatomically but also functionally map the output pathways of subpopulations of input-defined neurons."

Reviewer #1 (Remarks to the Author, italicized):

Feng Xiong et al present a new experimental approach for anterograde neurotracing with HSV129 vectors that reduces the experimental timeline from several weeks to under one week. The authors claim that their amplicon helper virus system overcomes several of the problems associated with existing approaches of anterograde labeling that utilize HSV namely low labeling intensity and high toxicity in postsynaptic neurons. In general the data do support this claim and demonstrate the potential of the amplicon system. The idea behind this new approach is pretty clever and the amplicon system could have a significant impact in the field even though it does not solve the problem of high toxicity in 1st order neurons.

(1) However, it is my understanding that 1st order neurons should be either green or double-labeled as presented in figure 1c. In practice the authors observe numerous 1st order neurons that are labeled brightly red (Fig.2c). This suggests that the system does not always work as it is presented in Figure 1c.

Reply (1): We thank the reviewer for their acknowledgment of the improvements conferred by our novel system and for allowing us the opportunity to present our new tracer system with greater clarity. H129_{Amp} tracer system consists of two components: the tracer and the helper. After the H129_{Amp} tracer system is simultaneously injected into the brain, the local neurons may be infected by the following potential combination of components: i) tracer alone (green), ii) helper alone (red), and iii) both the tracer and helper (yellow, Reply Fig. 1). The following description takes the H129_{Amp}-CTG tracer as a representative example: i) Neurons that are infected by the tracer alone are labeled green by GFP. The H129_{Amp} tracer, represented by H129_{Amp}-CTG, carries a pseudogenome formed by ~14-copy of concatemeric expression cassette of Cre-TK-GFP (CTG). The H129_{Amp} tracer contains no viral genes (except *TK*), and cannot replicate in the neurons by itself. But the 14-expression cassettes of CTG simultaneously and constitutively express GFP. Therefore, the cells infected by the tracer alone are labeled green. For these tracer alone infected neurons, there is no virus progeny produced in these neurons and no virus spread occurs. ii) Neurons that are infected by the helper alone are labeled red by tdTomato. The helper H129-dTK-T2-*pac*^{Flox} is thymidine kinase gene (*dTK*) deleted, which severely impaired its replication in neurons. The 2-copy tdTomato expression cassettes (T2) of the helper consistently express and accumulate the fluorescent protein tdTomato, thus labeling the cells red as shown in manuscript Fig. 2c. Again, for these helper alone infected neurons, there is also no virus progeny produced in these neurons and no virus spread occurs. iii) Neurons that are coinfecting by both the tracer and the helper are labeled yellow by both GFP and tdTomato. As described in the manuscript, co-infected tracer and helper obligatorily support each other to produce viral progeny, and the expression of tdTomato and GFP co-labeled the neurons yellow. The progenies of H129_{Amp}-CTG tracer are produced in these starter neurons, and further transmit to the postsynaptic neurons and label them green. Only these co-infected neurons are the potential starter cells for anterograde monosynaptic tracing using the H129_{Amp} tracer system.

Figure 2c in the manuscript shows the injection site at Day 1 after the tracer/helper simultaneous injection. The green, red, and yellow cells represent the neurons infected by tracer alone, helper

alone, and both tracer and helper, respectively, as described above.

The corresponding explanation and description have been added in the revised manuscript (line 141-144).

Reply Fig. 1: Schematic infection and labeling at the tracer system injection site. Elements of this can be found in the revised Fig. 1e in the manuscript.

(2) *The field has seen other ingenious approaches published in high impact journals that looked quite promising but have not found widespread use presumably because of experimental difficulties and problems with reproducibility. To allow a better assessment of the amplicon system in particular with regards to its reproducibility the authors should present a summary of the number and percentage of double labeled cells from several experiments like the one shown in Fig. 2c.*

Reply (2): We agree with the reviewer that this is an important piece of information. We performed quantitative analysis of the labeled AC neurons. An average of 588 ± 103 (means \pm SEM) yellow neurons (coinfecting by both tracer and helper), 352 ± 95 green neurons (infected by tracer alone), and 210 ± 56 red neurons (infected by helper alone) are observed in AC (injection site) (revised Fig. 2c). Double labeled cells account for 51% of total labeled cells, and green and red cells account for 31% and 18%, respectively. The result has been added in the revised manuscript as Fig. 2c, and described in the text accordingly (line 233- 236).

However we think that it is important to qualify this result by noting that the reproducibility of the co-infection efficiency is dependent on many variables, including the quality of the viral tracer, the ratio of tracer/ helper, and potentially the animal species and the specific circuit to be traced, and perhaps other factors that we have not yet identified. With this cautionary note in mind, we strongly suggest researchers perform test tracing and carefully optimize the component viral titers in relation to tracing quality when they are applying any kind of tracers, see also (PMID: 28499404, 32824837, 33367996, and 35012591). Experimental consistency is an important issue to be concerned about for viral tracer application and was indeed one of our primary motivations to work out novel strategies to develop the present H129_{Amp} tracer system with its streamlined single-injection protocol rather than the conventional sequential dual-injection protocol.

(3) *I would also like to see some data on the robustness of this system that is how sensitive it is to changes in the Amplicon/Helper ratio.*

Reply (3): The reviewer highlights the important issue of different tracer/helper ratios as they impact tracing robustness. In the original manuscript, we depicted our optimized condition with few details on how we arrived at that optimization. We now show the conditions that we tested and present these in Supplementary Fig. 1 to give the reader a better sense of how to design similar optimization protocols for their own applications.

To optimize tracing outcomes, we tested a range of tracer/helper ratios, including 10:1, 5:1, 1:1, 1:5, and 1:10 (Supplementary Fig. 1a). H129_{Amp} tracer (represented by H129_{Amp}-CTG) and H129-dTK-T2-*pac*^{Flox} helper (the only one, shared in all the experiments) were injected into AC of wildtype C57BL/6 mice as the indicated ratio, and the results were observed at Day 5. For tracer/helper ratios of 10:1 and 1:10, no neurons at the postsynaptic nuclei (represented by Cont. AC) were labeled by GFP. When tracer and helper were applied at ratios of 5:1, 1:5, and 1:1, GFP-labeled neurons were observed at Cont. AC, a representative postsynaptic nucleus. Quantification analysis showed that most GFP-labeled neurons were observed in Cont. AC with a 1:1 tracer/helper ratio (Supplementary Fig. 1b and c). These results suggest that tracer/helper at 1:1 ratio has the best tracing effect, and therefore is applied in all further tracing experiments.

These results have been added to the revised manuscript as Supplementary Fig. 1.

Reviewer #2 (Remarks to the Author):

This manuscript describes the development and testing of several new reagents for anterograde trans-neuronal tracing. While the results presented clearly show that the new reagents will have utility for experiments that are not possible with other existing tracers, there are many concerns about various claims that are made without experimental support, and/or potential limitations or labeling artifacts that the authors do not consider. The manuscript would also benefit considerably from clearer descriptions of the various reagents and how they are expected to behave and interact with each other. (The Overview below provides an example of how to do this.) This review begins by providing an overview of the various reagents used and their apparent intended uses. Then the review turns to describing possible limitations to the methods that the authors either fail to consider or claim to have considered but do not support with definitive evidence. The review then turns to problems with organization of the manuscript or lack of detail in descriptions that make it difficult to follow and which could be remedied by some re-writing. Various edits are suggested in the context of the original text. Finally, there are several minor points that could be addressed.

Overview of the reagents described and their intended uses:

The manuscript describes three different tracing systems. All three have in common that they utilize an HSV-H129 strain helper virus to allow the production/packaging of an amplicon vector ("tracer") that then spreads anterogradely to postsynaptic neurons. The helper is called H129-dTK-T2-LoxN-pac-LoxN. dTK refers to deletion of TK from the genome. TK is required for viral replication in dividing cells, such as neurons. T2 refers to the insertion of genes for TdTomato such that infected cells will express TdTomato. LoxN-pac-LoxN refers to the fact that the pac sequence is floxed allowing for its removal following

Cre-recombination. This is relevant because pac is required for packaging of viral genomes into viral particles. There are two different tracers used in these experiments as well as two different H129 viruses. How each is used depends on experimental goals of the 3 different systems.

The first system is intended to identify the directly postsynaptic neurons downstream of neurons infected in a brain area of interest. It does not distinguish between outputs of different neuron types in the brain area of interest. Here the helper is H129-dTK-T2-LoxN-pac-LoxN and the tracer is H129ampCTG, which expresses Cre-recombinase, TK, and GFP. These are mixed together and co-injected into the region of interest. In neurons that are co-infected, the helper genome is recombined by Cre in order to remove pac and prevent trans-neuronal spread of the helper. Expression of TK allows the helper to generate all of the required viral structural components and package the tracer genome into amplicons which are then transmitted down the axons and trans-neuronally to downstream recipients. GFP expression from the tracer then marks the downstream neurons. Insofar as pac is removed from the helper genome, the helper will not spread beyond the initially infected neurons. Furthermore, the tracer is unable to spread beyond the first synaptic step because it is unable to replicate in the absence of helper. The absence of helper from the downstream neurons is also expected to mitigate toxicity. GFP expression from the amplicon tracer is expected to be transient (cite references). If this approach is employed in a cre-reporter mouse line, such as AI14, expression of Cre from the amplicon tracer can permanently label downstream neurons.

The second system is intended to identify the directly postsynaptic neurons downstream of a selected population of neurons that express cre-recombinase (from a mouse driver line) in a brain area of interest. It is intended that the outputs of neighboring cre-negative neurons in the region of interest will not have their outputs labeled (but see below). For these experiments, the same helper is used (H129-dTK-T2-LoxN-pac-LoxN) but the amplicon (H129amp-DIO-TG) expresses only TK and GFP (not Cre), and their expression requires Cre-recombination. When this helper and tracer are co-injected in an area of interest of a cre driver line, pac will be removed from the helper genome in Cre+ cells, preventing it from spreading. Cre will also recombine the amplicon, allowing it to express TK such that the helper can replicate and facilitate production of tracer amplicons that spread to and express GFP in downstream neurons. Cre-negative neurons infected with helper and tracer will initially lack TK and will therefore not be able to produce amplicon tracer. Therefore, the downstream neurons of Cre- neurons would not be labeled. Note however, that for brain regions in which Cre-positive and Cre-negative neurons are interconnected (such as auditory cortex), labeling of downstream neurons will not actually be selective for those connected to the Cre-expressing neurons, as intended. This is because TK expressing amplicons will spread to Cre-negative neurons allowing the helper to be active and produce tracer amplicons in the Cre-negative neurons. See further below.

The third system is intended to trace circuits anterogradely across two synaptic steps. It uses the first system described above (H129-dTK-T2-LoxN-pac-LoxN helper and

H129ampCTG tracer) to initiate transfer of the tracer to downstream targets. In addition, a second H129-based viral tracer that was described in a previous publication (H129-dgK-G4) is injected into a downstream region. H129-dgK-G4 has an envelope protein required for packaging infectious articles deleted and expresses GFP. It can be transcomplemented by expression of gK to mediate monosynaptic anterograde spread. In order to trace connections across a second synaptic step an AAV helper virus with Cre-dependent gK is injected into the downstream region of interest at the same time that the H129-dTK-T2-LoxN-pac-LoxN helper and H129ampCTG tracer are injected. The H129-dgK-G4 is injected 3 weeks later. In the downstream target region H129ampCTG tracer that has spread to postsynaptic neurons expresses Cre and recombines the AAV helper. The resulting gK expression allows the H129-dgK-G4 to spread an additional synaptic step resulting in GFP label.

Possible limitations and labeling artifacts that are not considered or for which claims against them are not supported:

The description above describes how it is hoped that the tracing systems will work. But there are many possible scenarios not considered by the authors or not ruled out by results presented, that would result in very different undesired outcomes. Below, these are each considered in turn with a description of the relevant reagents and experiments and data presented that are relevant.

1: For tracing system 1, it is claimed that helper virus does not spread from the initially infected neurons and that, as a consequence, the spread of tracer is monosynaptic (not multi-synaptic spread). The validity of this claim relies on the unproven assumption that Cre-recombination from the tracer will successfully recombine and remove pac from every copy of the helper genome. This seems unlikely given that the helper will rapidly replicate in the presence of TK expressed from the tracer. (1) To support these claims it is necessary to show that there is no TdTomato expressed at postsynaptic target regions at very short times (1 or 2 days) after injection of the tracer system. For example, following injections of the tracing system into AC, please show photographs of the MG at d1 and d2 post-injection. Are there any red cells observed? Photographs at d5 are not adequate because it is clear that all cells infected with helper would have been killed and eliminated at d5. This is clearly shown in Fig. 2d which illustrates that red fluorescent cells have disappeared from the AC between d1 and d5. (Compare to Fig. 1c.)

Reply 1-(1): We agree that this is an important concern. As we noted in the revised Methods and manuscript, Cre-excision efficiency is very high – but we acknowledge the possibility that not every *pac* sequence will be removed from every copy of the helper genome. In anticipation of this potential issue, a very small fraction of helper was also packaged during H129_{Amp} tracer production in Vero cells, resulting in the raw H129_{Amp} tracer product as a mixture of tracer (~95%) and helper (~5%).

Upon brain injection of the H129_{Amp}-CTG tracer system, potentially incomplete *pac* excision could cause replication and transmission of the helper in theory. However, we do not observe helper labeled neurons in the downstream target of the injection site at Day 5 (Supplementary Fig.

4). As suggested by the reviewer, we tested this more comprehensively by examining the injection site (AC) and its innervating regions (represented by MG) at earlier time points after injection but before Day 5 post injection. No tdTomato labeled neurons are observed in postsynaptic target regions at either Day 1 to Day 3 post injection, indicating that the strategy that we employed for H129_{Amp} tracer production in Vero cells is sound (Reply Fig. 2).

These results indicated that the helper actually did not transmit to any of the downstream nuclei, despite the acknowledged theoretical possibility of helper-replication due to the Cre-recombination efficiency. We do not know entirely why this works, but we hypothesize the following potential reasons may contribute: i) The replication of H129 in neurons *in vivo* is slower and less efficient than that in the Vero cells used for H129_{Amp} tracer production. It is possible that the Cre-recombinase rapidly expressed by H129_{Amp} efficiently excises *pac* in every copy of the helper, and thus fully disarms the helper's packaging ability in neurons. ii) The H129_{Amp} tracer carries multiple copies of *pac* in the pseudo-genome, while the helper has only 1 copy before excision. Therefore, the replicated tracer may hijack the capsid packaging machinery for its own pseudo-genome, and the un-excised helper genome cannot be efficiently packaged under such conditions.

Reply Fig. 2: AC tracing results at early time points. Elements of this can be found in the revised Fig. 2c in the manuscript and Supplementary Fig. 4.

The H129_{Amp} tracer system (H129_{Amp}-CTG tracer 1.5×10^8 pfu/ml and helper 1.5×10^8 pfu/ml, in 300 nl) was injected into the auditory cortex (AC, AP: -2.80 mm; ML: -4.13 mm; DV: -2.38 mm) of wildtype C57BL/6 mice, and the brains were collected at 1, 2 and 3 days post-injection for imaging. The representative images of the injection site AC and the direct innervating regions, represented by MG are shown. Images with higher magnifications of the boxed areas are presented in the right

panels.

(2) If there are red cells in postsynaptic targets of the AC at d1, then it is clear that they could mediate spread of tracer another synaptic step. Please show results that clearly illustrate a lack of spread in cases where it might be expected. For example, the results state: "... but not in further downstream second order connected nuclei (data not shown). These results show that H129Amp tracer anterogradely labels directly connected postsynaptic neurons limited to monosynaptic connections." It is not possible to show something with results that are not shown. Please specifically state what structures were observed that might have potentially been labeled by multisynaptic spread. Such structures should contain the GFP-labeled axon terminals of neurons that were trans-neuronally labeled with the tracer. But they should not contain labeled postsynaptic neurons. Please provide photographs illustrating the presence of axon terminals in these locations without postsynaptic labeling.

Reply 1-(2): We apologize for not included the control data showing the absence of multisynaptic spread to downstream brain regions. We agree that this is an important control. The requested data is now presented in Supplementary Figure 5.

Auditory cortex (AC) directly innervates lateral amygdala (LA) and contralateral AC (Cont. AC), these brain areas in turn project to central amygdaloid nucleus (CeA) and contralateral medial geniculate nucleus (Cont. MG), respectively (Supplementary Figure 5a). After injecting H129_{Amp}-CTG tracer system into the AC of wildtype C57BL/6 mice, we examined CeA and Mont. MG, the representative 2nd-order innervating nuclei of AC, at Day 5 (Supplementary Figure 5b). Neither green (GFP) nor red (tdT) labeled cell bodies were observed in these regions (Supplementary Figure 5c and d), thus showing that uncontrolled 2nd order neuronal labeling does not occur.

Further, the H129_{Amp} tracer system was tested in Ai14 reporter mice, Cre expressed by H129_{Amp}-CTG could drive long-term and robust expression of Cre-dependent red fluorescence reporter. After injecting H129_{Amp}-CTG tracer system into the AC of Ai14 reporter mice, we examined the representative 2nd-order innervating nuclei of AC, CeA and Cont.MG, at Day14 (Supplementary Figure 5e). No red labeled cell bodies were observed in these regions, while a small amount red axon terminals was observed (Supplementary Figure 5f and g).

These results have been added to the revised manuscript as Supplementary Fig. 5.

2: The authors do not consider the known fact that both the helper and the tracer are packaged with envelope proteins that will mediate retrograde infection. The H-129 strain is biased toward anterograde because it is trafficked down the axons of infected neurons. But there is not any mechanism that prevents infection of axon terminals and retrograde spread to the infected cell bodies. (1) As a result, it is expected that injections of the tracing system into cortical locations (such as AC) will result in direct infection of thalamic neurons (such as in MG). This should be clearly visible as TdTomato labeling in the MG following injections of either tracer or helper alone (or both together) into AC. Please show evidence that none of these occurs. For example, show photographs of the MG at day 1 after injections of the tracing system into AC. Are there any red neurons in the MG. Again, note

that the absence of red label in MG at day 5 is not sufficient because directly infected neurons are clearly dead and disappear by d5. (2) Please also show evidence that tracer does not retrogradely infect and express GFP in the absence of helper. This is a crucial control. But surprisingly there are no data presented in the manuscript showing the nature of direct infection with the tracer in the absence of helper. Please show photos of the AC and other pre-and postsynaptic targets of AC (e.g. SC, MG, contralateral AC) following injections of tracer without helper.

Reply 2-(1): We appreciate the reviewer raising this essential concern about the potential retrograde labeling of H129 tracer. We are indeed aware of this issue and we have taken steps to mitigate this as detailed below.

i) The retrograde labeling of H129 tracers can be minimized by optimizing the tracing parameters. Despite the predominant anterograde transneuronal transmission, H129 tracers have the potential to retrogradely label the upstream neurons by invading the axon terminal and retrograde transportation (PMID: 24585022, 31348990). Our lab has also intensively investigated this retrograde labeling, and discussed this issue in our previous publications (PMID: 28499404, 32824837, 3336799, 35012591). According to our published studies, the retrograde labeling of H129 tracer is associated with many experimental parameters, such as tracer titer, injection volume, tracing duration, injecting site, the circuit to be traced, etc. Carefully optimizing these tracing conditions may effectively minimize or even prevent the potential retrograde labeling (PMID: 28499404). We have tested multiple doses of the H129 tracer in multiple brain regions and from these optimization studies, we determined that the ideal dose of the H129 tracer is $\sim 5.0 \times 10^8$ pfu/ml, 150-350nl volume. Using this optimized dose, H129 tracer labels no upstream neurons retrogradely for most tested brain nuclei, except for very few retrogradely labelled cells in CA1 (see Supp. Fig. 8 in Zeng *et al.*, 2017, Mol. Degeneration).

The H129_{Amp} tracer system applied in the present study are all at a dose of $\sim 1.5 \times 10^8$ pfu/ml, 300nl, which is below the threshold $\sim 5.0 \times 10^8$ pfu/ml, 350nl.

ii) No retrograde labeling was observed in the test tracing using the V1-SC pathway using the H129_{Amp} tracer system. The V1-SC pathway has been well characterized, and all available evidence indicates that it is unidirectional (PMID: 27989459). In the present study, we performed a test tracing using V1-SC pathway to validate the transmission direction of the H129_{Amp} tracer system. H129_{Amp} tracer system was injected into the SC of Ai14 mice with a dose ($\sim 1.5 \times 10^8$ pfu/ml, 300nl) lower than the above-mentioned threshold ($\sim 5.0 \times 10^8$ pfu/ml, 350nl). At Day 14, robust red labeled cell bodies were observed in postsynaptic targets of SC (Supplementary Fig. 6c-g), while no labeled cell bodies in upstream region V1 are detected (Supplementary Fig. 6d-g). These results indicated that the H129_{Amp} tracer system does not retrogradely label the upstream nuclei under the optimized condition.

iii) The retrograde labeling of H129 tracers can be significantly reduced using gK_{mut}, and we are working on further improvements of the H129_{Amp} tracer system. In a recently published paper, we introduced a novel H129 tracer with reduced retrograde labeling (PMID: 35012591). This is achieved by pseudotyping H129 tracer with the mutant gK (gK_{mut}), an envelope glycoprotein of H129 (see Fig. 4 in Yang *et al.*, 2022, Mol. Degeneration). Currently, we are working on replacing

the original wildtype gK (gK_{wt}) with the gK_{mut} in the helper genome. We believe the gK_{mut} replacement should also dramatically reduce the unwanted retrograde labeling of the H129_{Amp} tracer system.

Reply 2- (2): We tested the transmission ability of the tracer (H129_{Amp}-CTG) without adding additional helper. The raw product of H129_{Amp}-CTG tracer was titrated and adjusted to the optimized tracing titer (1.5×10^8 pfu/ml, *This is considered as “tracer alone”, since a very small fraction of helper “contamination” is inevitable and irremovable in raw H129_{Amp} tracer product*). Without additional helper supplementation, the H129_{Amp}-CTG tracer was injected “alone” into the AC of wildtype C57BL/6 mice, and the brains were examined at Day 1 and Day 5. Green (GFP) labeled neurons were clearly observed at the injection site, but not detected in any other examined brain regions, including the upstream and postsynaptic target regions of AC (Supplementary Figure 2a and b). These results indicate that the tracer alone without additional helper is limited to the injection site.

Notably, the derived H129_{Amp} tracer is a mixture containing ~5% helper, as we described in the manuscript and methods. It is hardly to obtain pure high titer H129_{Amp} tracer without any helper “contamination” (PMID: 19956558). The titers of the tracer and helper in the raw tracer product mixture can be determined by plaque-forming assays, and then independently propagated helper needs to be added to the raw tracer product to adjust the tracer/helper to the desired final ratio. In Supplementary Figure 1, we tested the spreading ability of tracer/helper ratio of 10:1, 5:1, 1:1, 1:5, and 1:10. H129_{Amp} tracer (represented by H129_{Amp}-CTG) and H129-dTK-T2-*pac*^{Flox} helper (the only one, shared in all the experiments) were injected into AC of wildtype C57BL/6 mice as the indicated ratio, and the results were observed at Day 5. For tracer/helper ratios of 10:1 and 1:10, no neurons at the postsynaptic nuclei (represented by Cont. AC) were labeled by GFP. When tracer and helper were applied at ratios of 5:1, 1:5, and 1:1, GFP-labeled neurons were observed at Cont. AC, a representative postsynaptic nucleus. Quantification analysis showed that most GFP-labeled neurons were observed in Cont. AC with a 1:1 tracer/helper ratio (Supplementary Fig. 1b and c). The results indicated that the H129_{Amp} tracer system didn't spread when the absolute ratio between tracer and helper is too big (exceed 5), which is a similar condition to the raw tracer product (tracer ~95% : helper ~5% = 19:1).

These results have been added to the revised manuscript as part of the Supplementary Fig. 2.

3. The second “starter-specific” tracing system is unlikely to actually be specific as intended and claimed. This system utilizes a Cre-dependent tracer (H129Amp- DIO-TG) and is intended to trace the monosynaptic outputs selectively from Cre-expressing cells. The authors do not describe the rationale behind how this should work, but apparently they expect that there will not be any expression of TK in cells that do not express Cre. And since Cre-negative cells at the primary injection site are infected with helper but do not express TK, the helper will not replicate in Cre-negative cells. It is also assumed that tracer will not spread locally between Cre-positive and Cre-negative cells. But these assumptions are certainly wrong for the cortical experiment shown. This is because there are direct connections between the Cre-positive and the Cre-negative cortical cells. It is therefore unlikely that the postsynaptic label observed in distant structures reflects selective outputs

from the Cre-positive cortical cells. The expected mechanism for non-selective spread is as follows. Following injections of the tracer and helper into the cortex, both vectors infect both Cre⁺ and Cre⁻ neurons. In the Cre⁺ neurons, pac is removed from the helper and the tracer expresses TK and GFP. pac is not removed from the helper that is in the Cre⁻ neurons. Tracer spreads from the Cre⁺ to connected Cre⁻ neurons where it expresses TK and allows replication of the helper and further production of tracer amplicons in the Cre⁻ neurons. This allows the tracer to spread an additional synaptic step and to label the outputs of Cre⁻ neurons. In addition, helper with intact pac is also replicated and can spread from the Cre⁻ neurons, resulting in further multisynaptic spread.

The authors do not appear to have considered this and did not use a Cre-expressing mouse line that is capable of demonstrating this artifact. For example, it is expected that if the experiment were conducted in the AC of PV-Cre mice expressing Cre in parvalbumin inhibitory neurons (which make only local cortical connections) “starter-specific” tracing would not result in spread outside the cortex. But the artifact described would allow the spread of tracer from PV interneurons to cortical excitatory neurons, and then from the excitatory cells to postsynaptic neurons in more distant targets. It is necessary to show results from a Cre line in which it is possible for this artifact to be revealed and evaluated. Another such Cre line would be Tlx3-Cre which expresses Cre in cortical layer 5 pyramidal neurons that make cortico-cortical projections but not those that project to subcortical structures. If the spread is actually “starter-specific”, use of the system in this line should result in anterograde tracing to cortical targets but no labeling in subcortical structures such as superior colliculus.

Reply 3: We appreciate the reviewer raising this essential concern and offering the experimental strategy to address it.

The reviewer describes the possible chain of events that: i) the H129_{Amp}-DIO-TG tracer and helper infect the same Cre⁺ neurons at the injection site; ii) the tracer and helper support each other and new tracer progeny is synthesized, which is not H129_{Amp}-DIO-TG anymore, but consistently expresses TG (similar to H129_{Amp}-TG), because of Cre-induced recombination; iii) this newly synthesized tracer H129_{Amp}-TG may be anterogradely transmitted to local Cre⁻ neuron innervated by the starter Cre⁺ neuron; iv) if this local Cre⁻ neuron is also infected by helper, the H129_{Amp}-TG and helper may support each other independently to Cre-recombinase and initiate further anterograde transmission again through this Cre⁻ neuron. We do agree with the reviewer that if this chain of events were to occur, it would complicate the interpretation for the starter-specific tracing.

To address the reviewer’s concern, we performed tracing with H129_{Amp}-DIO-TG tracer system in PV-Cre mice, instead of the Tlx3-Cre mice (which are unavailable for us). PV-Cre mice express Cre recombinase in parvalbumin inhibitory interneurons (making only local cortical connections). The H129_{Amp}-DIO-TG tracer system was injected into the AC of the PV-Cre mice, and the results were examined at Day 5. As shown in Supplementary Fig.11, the green (GFP) labeled neurons are only observed at the injection site, but are not detected in any other brain regions, including known downstream and upstream regions.

Thus while the H129_{Amp}-DIO-TG tracer system in theory could initiate the anterograde

monosynaptic transmission from a local Cre⁻ neuron in the chain of hypothetical events described above, this is not detected in practice. The most likely explanation for the absence of Cre⁻ neuron initiated transmission is that it may be a very low probability event. However, this potential caveat should be taken into account when applying the H129_{Amp}-DIO-TG tracer system for starter-specific tracing. To warn future users of this hypothetical problem, we have added text addressing this in the revised manuscript along with our new control data (lines 491-499, and Supplementary Fig.11).

Other comments/suggestions:

1. *“Although multiple anterograde monosynaptic tracers have been developed based on the herpes simplex virus 1 (HSV-1) strain H129 (H129), functional mapping for output connectome requires further technical development.” Suggest changing to “...connectome would benefit from further ...”.*

Reply: We concur and have amended the revised manuscript to “Although multiple anterograde monosynaptic tracers have been developed based on the herpes simplex virus 1 (HSV-1) strain H129 (H129), functional mapping for output connectome requires further technical development to minimize viral toxicity” (lines 78-81).

2. *“Both retrograde and anterograde monosynaptic tracing are achieved by combining two components: i) a viral tracer (tracer), which is a replication- or transmission-deficient virus with one or more genes deleted whose trans-neuronally infects connected neurons requires the assistance of ii) a helper virus (helper) which complementarily expresses the tracer’s deficient gene(s), thus supporting tracer replication and transmission.” Suggest the following edits: “Both retrograde and anterograde monosynaptic tracing are achieved by combining two components: i) a conditionally competent viral tracer (tracer) and ii) a helper virus (helper). The viral tracer is replication- and/or transmission deficient due to the deletion of one or more genes that are required for transneuronal spread to connected neurons. The helper virus expresses genes which replace or complement tracer’s deficient gene(s), thus supporting tracer replication and/or transmission.”*

Reply: We concur and have amended the revised manuscript to “Retrograde and anterograde monosynaptic tracing can be achieved by combining two components: i) a conditionally competent viral tracer (tracer) and ii) a helper virus (helper). Replication- and/or transmission deficiencies due to the deletion of one or more genes that are required for transneuronal spread to connected neurons of the viral tracer component is a feature that enforces monosynaptic spread. The helper virus component expresses genes that complement the tracer’s deficient gene(s), thus supporting tracer replication and/or transmission” (lines 81-87).

The following sentence is factually incorrect and needs to be changed. “The AAV helper has limited expression efficiency due to a single copy of target gene and deficiency in

replication, thus needs to be administrated first of two separate injections to allow sufficient target expression and accumulation to support replication and transmission of tracer.” AAV genomes concatenate to incorporate many gene copies into episomes. Thus, AAV is not limited to a single copy, and is capable of expressing at very high levels. It is more relevant to this manuscript that expression is slow and that AAV has a limited payload. Suggest changing to: “Due to the slow onset of expression for AAV vectors, the use of AAV necessitates two separate injections, first helper and then tracer, to allow sufficient expression and accumulation to support replication and transmission of the tracer.”

Reply: We concur and have amended the revised manuscript to “Due to the slow onset of expression for AAV vectors, the use of AAV necessitates two separate injections, first helper and then tracer, to allow sufficient expression and accumulation of the complementary gene and to support the tracer’s replication and transmission” (lines 90-93).

3. The introduction provides no context for the statement: “Moreover, for the first time, input-defined postsynaptic neurons’ monosynaptic anterograde tracing is succeeded by combining H129Amp tracer system with H129-dgK-G4 tracer.” (1)For the reader to understand this statement it will be necessary to describe the H129-dgK-G4 tracer. (2)What are the expected advantages (if any) of using H129-dgK-G4 tracer with H129amp tracer? (3)How is it possible to combine these for cell type specific output labeling since H129-dgK-G4 tracer expresses pac and is not disabled by cre-dependent pac removal?

Reply: We can clarify this.

(1) H129-dgK-G4 is an anterograde monosynaptic tracer developed by our laboratory and introduced in a recent publication (PMID: 35012591). It is a gK gene deleted (dgK) recombinant virus with 4 copies of the GFP coding sequence (G4). When infecting neurons alone, H129-dgK-G4 expresses GFP and intensively labels the cells, but it does not spread among neurons due to the lack of gK for virus shedding. *In trans* expressed gK by AAV complementarily supports H129-dgK-G4 anterograde transmit the downstream target.

(2) Current anterograde tracers achieve monosynaptic 2-order neuronal tracing (A to B), but fail to achieve 3-order tracing. The combination of H129_{Amp} tracer system and H129-dgK-G4 tracer system makes it possible to perform trans-2-synaptic 3-order anterograde tracing (A to B to C).

(3) We assume the reviewer raises this question for the potential application shown in Supplementary Fig. 10, where we have raised 2 different tracing scenarios: A (Cre⁺) to B to C, and A to B (Cre⁺) to C.

Similar to the input-defined output tracing as shown in Fig.4, these cell type-specific tracing examples also apply H129-dgK-G4 and appropriate AAV helper for the second step transmission. In both tracing scenarios, the appropriate H129_{Amp} tracer system was injected into the 1st-order group of neurons, and then the H129_{Amp} tracer transmits to the 2nd-order neurons either in a Cre-dependent (Supplementary Fig. 10a) or -independent (Supplementary Fig. 10b-c) manner. In the 2nd-neurons, the H129_{Amp} expresses Flp-recombinase, and initiates the gK expression of the AAV-

helper. When H129-dgK-G4 is injected into the 2nd-order region at Day 21, it propagates upon the assistance of the accumulated gK, and the newly produced H129-dgK-G4 transmitted to and labeled the 3rd-order neurons. Notably, at the time when H129-dgK-G4 is administered, the H129_{Amp} transmission to the 2nd-order neurons has been eliminated as shown in Supplementary Fig. 4. H129_{Amp} and H129-dgK-G4 will not support each other's replication in the 2nd-order neurons. Therefore, the pac-competence of H129-dgK-G4 is required for successful packaging and transmission for the 2-step tracing.

4. There is nothing in the Introduction or the Results description of generation, production and tracing principles to make it clear that, for the first tracing system described, the initial infection or production of tracer is not cell type specific. Please state clearly that the initial injection of tracer and helper will infect and spread from all types of neurons at the injection site.

Reply: We concur that this could be more clear.

As commented by the reviewer, the first H129_{Amp} tracer system (H129_{Amp}-CTG tracer and H129-dTK-T2-*pac*^{Flox} helper) initiates anterograde transmission from any cell type, and is not capable to perform starter-specific tracing.

We have amended the revised manuscript to make this more clear (lines 215-217).

5. "Abundant AC neurons were labeled by H129Amp tracer (green) and/or helper (red)." Why are neurons labeled red? Is there a red fluorescent protein expressed from the helper genome? Is the tissue stained with an antibody against the helper? There is nothing in the introduction or the results description of generation, production and tracing principles to indicate that TdTomato is expressed from the helper genome. This can only be guessed by looking at Figure 1b. There is also nothing in the figure 1 legend to indicate that this guess is correct. The following edits are suggested. At line 125 rewrite as: "The H129Amp tracer system is composed of H129Amp tracer and H129-dTK-T2-LoxN-pac-LoxN helper. H129-dTK-T2-LoxN-pac-LoxN helper has the thymidine kinase gene deleted (dTK), expresses two copies of the fluorescent protein tdTomato (T2), and its pac gene can be deleted by Cre-recombination (LoxN-pac-LoxN). Due to the identical infection features of H129amp tracer and helper, the system it requires only a single-injection instead of the conventional sequential twice-injection." Also update the figure 1 legend to clearly indicate what is schematized in 1b.

Reply: We concur that this could have been more clear and have amended the revised manuscript (lines 140-151).

6. The ED Fig. 2 legend and the figure refer to LP. Based on the photograph it appears that the location indicated by LP is the lateral pulvinar nucleus of the thalamus, which receives direct input from SC. But the figure legend states: "LP, pretectal area;". This is

probably wrong.

Reply: We concur and have corrected this in the revised manuscript (Supplementary Fig. 6).

7. “These results show that the postsynaptic neurons labeled by H129Amp tracer system maintain normal physiological conditions, thus allowing for functional connection mapping.” This result cannot be generalized to all neurons labeled with the tracer. This only applies to cells that are still present after 14 days. Other cells may have died.

Reply: When transmitting to the postsynaptic neurons, H129_{Amp} neither replicates nor expresses any toxic viral proteins, so it doesn't kill cells as other replicable viruses do. However, H129_{Amp} might also induce immune responses due to the carried-over virion proteins and the bacterial DNA sequences in its pseudo-genome. Based on these properties the H129_{Amp} tracer has very low or even no viral toxicity, however we concede that it may still cause mild damage to the cells, possibly due to cell stress or immune responses. We acknowledge this possibility and have amended the text to “Patch clamp analysis of postsynaptic neurons labeled by the H129_{Amp} tracer system indicates that these cells maintain normal physiological conditions and may be thus useful for functional connection mapping. However at present, we cannot rule out possible secondary effects of cell stress or immune responses that may impact yet to be determined aspects of cellular physiology.” in the revised manuscript (lines 295-299).

8. “Notably, H129Amp tracer system is also capable of achieving input-defined outputs mapping from specific starter neurons, which cannot be done using transsynaptic AAV2/1.” There is no reason to expect that this could not be done with transsynaptic AAV2/1. If AAV2/1 expressing Cre is used in a mouse with Cre-conditional Chr2 expression (either from the genome or from an AAV-DIO-Chr2 injected at the postsynaptic site) then the same experiment could be done.

Reply: To the best of our knowledge, the AAV2/1 has no cell-specificity at the injection site. That is to say, AAV2/1 cannot perform the output tracing of a neuron, which is the innervating target of the Cre⁺ neurons. But this can be potentially achieved by the H129_{Amp} tracer as introduced in the manuscript (Supplementary Fig. 8).

However, we appreciate the reviewer's comments, and the corresponding expression has been modified to “Notably, H129_{Amp} tracer system is also applicable for input-defined output mapping from specific starter neurons.” (lines 306-307).

9. “Fast tracing. All current monosynaptic tracers, both retrograde and anterograde, use AAVs as helpers.” This statement is not true. For example, monosynaptic rabies tracing has been conducted using mouse lines that have cre-dependent or tTA-driven expression of the helper genes. Thus, there is no need for injection of AAV helper or to wait for expression of helper genes.

Reply: We apologize for this arbitrary statement. In the revised manuscript we have updated the text to reflect this (lines 402-403).

10. *"In addition to the single-copy target, deficiencies in the replication of viral genome and production of viral particle further limit the expression efficiency of AAV genetic payload." As noted previously, AAV is able to express its payload very efficiently and without toxicity. Replication is not necessary for efficient long-term expression from AAV vectors.*

Reply: We concur and have amended the revised manuscript (lines 403-405).

11. *"In the novel H129Amp tracer system, helper, the toxic viral protein producer, is left behind in the starter neurons." (1) As noted above, there are no data presented to support this assertion. It is plausible that the Cre expression from the tracer does not completely remove cap from all helper-infected neurons and that some helper spreads trans-neuronally, along with the tracer. Subsequently, the co-infected trans-neuronally labeled cells die. This is consistent with the observation that GFP expression does not persist in labeled postsynaptic neurons. (2) Please explain why GFP expression is not persistent. I was able to find this in papers cited but readers should not have to track down the relevant facts.*

Reply (1): This concern is similar to Question#1 of Major Concern#1.

As we introduced in the revised Methods and manuscript, the Cre-excision efficiency is not sufficient enough to excise the *LoxN-pac-Lox* from every copy of the helper. Therefore, a very small portion of helper was also packaged during H129_{Amp} production, resulting in the produced raw H129_{Amp} tracer as a resulting mixture of tracer (~95%) and helper (~5%).

When applying the H129_{Amp} tracer system, inadequate *pac* excision may theoretically cause the replication and transmission of the helper. However, we do not observe helper labeled neurons in the downstream target of the injection site at Day 5 (Supplementary Fig. 4). As suggested by the reviewer, we examined the injection site (AC) and its innervating regions (represented by MG) at earlier time points. As shown no TdTomato labeled neuron was observed in postsynaptic target regions even at Day 1 to Day 3 (Reply Fig. 2).

These results show that the helper does not transmit to any of the downstream nuclei, contrary to the hypothetical scenario outlined by the reviewer. We hypothesize possible reasons for this. i) The replication of H129 in the neurons *in vivo* is slower and much less efficient than in the Vero cells *in vitro*. Cre-recombinase rapidly and massively expressed by H129_{Amp} is adequate to efficiently excise the *pac*-sequence in the helper, thus disarming the helper's packaging ability. ii) The H129_{Amp} tracer carries multiple copies of *pac*-sequence in the pseudo-genome, while the helper has only 1 copy. Therefore, the replicated tracer might hijack the capsid packaging machinery for its own pseudo-genome, and the un-excised helper genome cannot be efficiently packaged.

Reply Fig. 2: AC tracing results at early time points. Elements of this can be found in the revised Fig. 2c in the manuscript and Supplementary Fig. 4.

The H129_{Amp} tracer system (H129_{Amp}-CTG tracer 1.5×10^8 pfu/ml and helper 1.5×10^8 pfu/ml, in 300 nl) was injected into the auditory cortex (AC, AP: -2.80 mm; ML: -4.13 mm; DV: -2.38 mm) of wildtype C57BL/6 mice, and the brains were collected at 1, 2 and 3 days post-injection for imaging. The representative images of the injection site AC and the direct innervating regions, represented by MG are shown. Images with higher magnifications of the boxed areas are presented in the right panels.

Reply (2): It has been reported that the transcription of genetic payload in HSV amplicon is silenced rapidly after the quick and massive expression (PMID: 19223864, 17653098). The exact mechanism behind this silencing remains to be fully elucidated, but the intrinsic immune response may be involved. When infecting cells, H129_{Amp} tracer neither replicates nor expresses any toxic viral proteins, so it doesn't kill cells as other replicable viruses do. However, we acknowledge that H129_{Amp} might also induce immune responses by the carried-over viral proteins and pseudo-genome. i) When the HSV amplicon infects cells, the infection process may potentially trigger immune responses including those involving type I interferons. These responses in turn could suppress transgene expression at the transcriptional level by activation of STAT1 (PMID: 19223864, PMID: 17653098).

ii) The H129_{Amp} particle carries a set of viral structural proteins. Upon H129_{Amp} entering cells, these carried-over viral proteins may be sufficient to evoke immune responses of unknown magnitude.

iii) The pseudo-genome carried by the H129_{Amp} contains some prokaryotic DNA sequences, which are from the amplicon plasmid. These bacterial DNA sequences could hypothetically modulate

transcriptional silencing of the entire vector sequence after infection (PMID: 16537596, PMID: 33869657).

To account for these possibilities, we have amended the text (lines 462-467) in the revised manuscript.

12. Recent improvements in HSV amplicon vectors might allow more stable gene expression. For example see Soukupova et al., 2021; Improvement of HSV-1 based amplicon vectors for a safe and long-lasting gene therapy in non-replicating cells. [https://www.cell.com/molecular-therapy-family/methods/fulltext/S2329-0501\(21\)00060-7](https://www.cell.com/molecular-therapy-family/methods/fulltext/S2329-0501(21)00060-7). This might be worth mentioning in the discussion.

Reply: We now include this citation and acknowledge that this is an important future direction.

Reviewer #3 (Remarks to the Author):

In this study, Xiong et al. developed a novel HSV-1-H129 amplicon tracer system as a monosynaptic viral tracer for dissecting neuronal connectomes and targeted delivery of molecular sensors and effectors. The improvements of the H129Amp tracer system shorten the experiment duration from 28-days to 5-days for fast-monosynaptic tracing and minimize toxicity in the postsynaptic neurons. The idea behind this study is very exciting. However, some of the experiment's design and interpretations need more data and or to be further clarified and developed.

Reply: We thank the reviewer for their clear overall enthusiasm for our work. We address the reviewer's remaining concerns as detailed below.

1. The authors stated: " For anterograde monosynaptic tracing, the tracer system was administrated into the brain region of interest by a single-injection. In neurons, H129Amp tracer does not replicate alone since it contains no viral gene, but it efficiently expresses Cre, TK, and GFP with its multiunit cassettes in the concatemeric pseudo-genome (Fig. 1d)." It is possible that the GFP can be present in other brain areas to a retrograde transfection mechanism and or transneuronally.

Reply: We agree with the reviewer that this is an important issue, which is similar to Question #1, Major Concerns #2 of Reviewer #2.

We are indeed aware of this issue as shown in our past publications, and we have taken steps to mitigate this as detailed below.

i) The retrograde labeling of H129 tracers can be minimized by optimizing the tracing parameters. Despite the predominant anterograde transneuronal transmission, H129 tracers have the potential to retrogradely label the upstream neurons by invading the axon terminal and retrograde transportation (PMID: 24585022, 31348990). Our lab has also intensively investigated this retrograde labeling, and discussed this issue in our previous publications (PMID: 28499404, 32824837, 3336799, 35012591). According to our published studies, the retrograde labeling of H129 tracer is

associated with many experimental parameters, such as tracer titer, injection volume, tracing duration, injecting site, the circuit to be traced, etc. Carefully optimizing these tracing conditions may effectively minimize or even prevent the potential retrograde labeling (PMID: 28499404). We have tested multiple doses of the H129 tracer in multiple brain regions and from these optimization studies, we determined that the ideal dose of the H129 tracer is $\sim 5.0 \times 10^8$ pfu/ml, 150-350nl volume. Using this optimized dose, H129 tracer labels no upstream neurons retrogradely for most tested brain nuclei, except for very few retrogradely labelled cells in CA1 (see Supp. Fig. 8 in Zeng *et al.*, 2017, Mol. Degeneration).

The H129_{Amp} tracer system applied in the present study are all at a dose of $\sim 1.5 \times 10^8$ pfu/ml, 300nl, which is below the threshold $\sim 5.0 \times 10^8$ pfu/ml, 350nl.

ii) No retrograde labeling was observed in the test tracing using the V1-SC pathway using the H129_{Amp} tracer system. The V1-SC pathway has been well characterized, and all available evidence indicates that it is unidirectional (PMID: 27989459). In the present study, we performed a test tracing using V1-SC pathway to validate the transmission direction of the H129_{Amp} tracer system. H129_{Amp} tracer system was injected into the SC of Ai14 mice with a dose ($\sim 1.5 \times 10^8$ pfu/ml, 300nl) lower than the above-mentioned threshold ($\sim 5.0 \times 10^8$ pfu/ml, 350nl). At Day 14, robust red labeled cell bodies were observed in postsynaptic targets of SC (Supplementary Fig. 6c-g), while no labeled cell bodies in upstream region V1 are detected (Supplementary Fig. 6d-g). These results indicated that the H129_{Amp} tracer system does not retrogradely label the upstream nuclei under the optimized condition.

iii) The retrograde labeling of H129 tracers can be significantly reduced using gK_{mut}, and we are working on further improvements of the H129_{Amp} tracer system. In a recently published paper, we introduced a novel H129 tracer with reduced retrograde labeling (PMID: 35012591). This is achieved by pseudotyping H129 tracer with the mutant gK (gK_{mut}), an envelope glycoprotein of H129 (see Fig. 4 in Yang *et al.*, 2022, Mol. Degeneration). Currently, we are working on replacing the original wildtype gK (gK_{wt}) with the gK_{mut} in the helper genome. We believe the gK_{mut} replacement should also dramatically reduce the unwanted retrograde labeling of the H129_{Amp} tracer system.

2. The authors should provide data images that include a single injection of H129Amp tracer, showing that the GFP expression is only visible in the injection site and not outside the injection site at the least Day7 and Day 21 post-injection.

Reply: We concur with the reviewer that this issue is important, and note that it is similar to Question #2, Major Concerns #2 of Reviewer #2.

As suggested by the reviewer, we tested the transmission ability of the tracer (H129_{Amp}-CTG) without adding additionally helper. The raw product of H129_{Amp}-CTG tracer was titrated and adjusted to the optimized tracing titer (1.5×10^8 pfu/ml, *This is considered as “tracer alone”, since a very small fraction of helper “contamination” is inevitable and irremovable in raw H129_{Amp} tracer product*). Without additional helper supplementation, the H129_{Amp}-CTG tracer was injected alone into the AC of wildtype C57BL/6 mice, and the brains were examined at Day 7 and Day 21.

At both time points, (Day 7 and Day 21), green (GFP) labeled neurons were not detected throughout the examined brain regions, including the injection site and postsynaptic target regions of AC (Supplementary Fig.2c and d). These new results have been added to the revised manuscript as part of the Supplementary Fig. 2. It has been reported that the transcription of genetic payload in HSV amplicon is silenced quickly after the rapid and massive expression, peaking at 1 day post injection (dpi) and then dropping by over 3 logs within a week (PMID: 33869657, 19223864, 17653098). In the present study, we also observed the target expression over time course, which is consistent with the reported quick silencing. The fluorescence labeling of the H129_{Amp} tracer system could be observed as early as 1 dpi (at the injection site), but this signal quickly dims below the detection threshold for confocal microscopy imaging at 7 dpi (Supplementary Fig.4).

Since the labeling of both H129_{Amp} tracer and helper are no longer detectable over Day 7, we also performed observations at earlier time points. At Day 1 and Day 5, green (GFP) labeled neurons are clearly observed at the injection site, but are not detected in any other examined brain regions, including known upstream and postsynaptic target regions of AC (Supplementary Fig.2a and b). These results indicated that the tracer alone without additional helper is restricted to the injection site.

Notably, the derived H129_{Amp} tracer is a mixture containing ~5% helper, as we described in the manuscript and methods. It is hardly to obtain pure high titer H129_{Amp} tracer without any helper “contamination” (PMID: 19956558). The titers of the tracer and helper in the raw tracer product mixture can be determined by plaque-forming assays, and then independently propagated helper needs to be added to the raw tracer product to adjust the tracer/helper to the desired final ratio. In Supplementary Figure 1, we tested the spreading ability of tracer/helper ratio of 10:1, 5:1, 1:1, 1:5, and 1:10. H129_{Amp} tracer (represented by H129_{Amp}-CTG) and H129-dTK-T2-*pac*^{Flox} helper (the only one, shared in all the experiments) were injected into AC of wildtype C57BL/6 mice as the indicated ratio, and the results were observed at Day 5. For tracer/helper ratios of 10:1 and 1:10, no neurons at the postsynaptic nuclei (represented by Cont. AC) were labeled by GFP. When tracer and helper were applied at ratios of 5:1, 1:5, and 1:1, GFP-labeled neurons were observed at Cont. AC, a representative postsynaptic nucleus. Quantification analysis showed that most GFP-labeled neurons were observed in Cont. AC with a 1:1 tracer/helper ratio (Supplementary Fig. 1b and c). The results indicated that the H129_{Amp} tracer system didn't spread when the absolute ratio between tracer and helper is too big (exceed 5), which is a similar condition to the raw tracer product (tracer ~95% : helper ~5% = 19:1).

These results have been added to the revised manuscript as part of the Supplementary Fig. 2.

3. The same experiments need to be repeated for the helper.

Reply: Following the Reviewer's suggestion, we have done additional experiments to address this control. We injected the helper (H129-dTK-T2-*pac*^{Flox}) alone into the AC of wildtype C57BL/6 mice, and brains were collected at Day 5, Day 7, and Day 21 for observation. tdT labeled neurons (red) are only observed at the injection site at Day 5, but not in other examined brain regions, including known postsynaptic targets of AC (Supplementary Fig. 3a).

At the later time points (Day 7 and Day 21), tdT labeled neurons were not detected in any examined brain regions (Supplementary Fig. 3b and c). The disappearance of the labeling signal over time at the injection site may be due to simple protein turnover or possibly helper expressed toxic viral protein which may have triggered immune responses, resulting in elimination of the infected neurons.

These results show that the helper alone does not spread, and these new results have been added to the revised manuscript as Supplementary Fig. 3.

4. Figure 2a and b provided by the authors make it impossible to determine that the virus was injected into the primary AC.

Reply: We concur that this was unclear. In the revised manuscript, we have updated the figure to show this clearly (revised Fig. 2).

5. Figure 2c. (1) What is the fraction of the co-transfected neurons compared to the single transfected and (2) in which layers of the AC are located? (3) The figure legend of this figure is very confusing. For example, it is not clear if the panels h1 and h2 are a high magnification of the pyramidal neurons of the contralateral AC.

Reply (1): We appreciate the reviewer raising this concern and this concern was noted by reviewer 1 as well.

Following the reviewer's comment, we performed quantitative analysis of the labeled AC neurons. An average of 588 ± 103 (means \pm SEM) yellow neurons (coinfecting by both tracer and helper), 352 ± 95 green neurons (infected by tracer alone), and 210 ± 56 red neurons (infected by helper alone) are observed in AC (injection site) (revised Fig. 2c). Double labeled cells account for 51% of total labeled cells, and green and red cells account for 31% and 18%, respectively. The result has been added in the revised manuscript as Fig. 2c.

Reply (2): In Fig. 2c, the tracer system was injected to AC using the following coordinates: AP -2.80 mm / ML -4.13 mm / DV -2.38 mm, which should be positioned Layer 3 to Layer 4. According to DAPI staining, we have briefly marked the layers, and the labeled neurons primarily located between Layer 2 and Layer 6 of AC (Reply Fig.3). But it is difficult to distinguish specific layers only based on the DAPI staining, so we didn't further quantify distribution of the labeled neuron in different layers.

Reply (3): We have revised the figure legend to be more clear.

Reply Fig. 3: Labeled neuron distribution in AC. This is only shown in point-by-point reply.

6. The panel 2h depicts the contralateral AC with only a few neurons. Can the authors provide the number of the neurons that are GFP positive in the Cont AC? This is extremely important for the experiments performed in the next section (Figure 3).

Reply: We concur that this issue is important.

Following the Reviewer's comment, we quantified the GFP+ neurons in Cont. AC at Day 5 in Fig. 2, as well as the mCh+ neurons at Day 21 in Fig. 3 (Reply Fig. 4a). At Day 5 in Fig. 2, an average of 740 ± 146 (means \pm SEM) GFP+ neurons are observed in Cont. AC directly. Fluorescent protein needs to accumulate to a certain amount/level to reach the detectable threshold for confocal microscopy imaging. Immunostaining can amplify the target signal. More GFP+ neurons are detected in Cont. AC after immunostaining to amplify the GFP signal (Reply Fig. 4b and c). These results indicated that some neurons infected by the H129_{Amp} tracer weren't detected because of inadequate GFP level. Therefore, the observed GFP+ neurons don't fully represent all neurons infected by transmitted H129_{Amp}, since neurons with very low level GFP protein is undetectable, and immunostaining can help to bring up GFP signal.

In addition, reporter system can also amplify signal. The reporter system applied in revised Fig. 3 addressed this problem, which has also been applied in many other systems including the transneuronal AAV2/1 tracer (PMID: 27989459). H129_{Amp}-CTG tracer anterogradely transmitted through one synapse from the injection site (L-AC) to and infected the target neurons in R-AC. There, it expresses Cre, which in turn triggers ChR2-mCh expression of AAV2/9-DIO-ChR2-mCh. The presence of Cre-recombinase of H129_{Amp}-CTG for a limited time is sufficient to catalyze recombination of AAV2/9-DIO-ChR2-mCh genome, and permanently turns it into a constitutive expression form. From then onwards, the expression of ChR2 and mCh is independent of Cre. Quantifying the results of this, we detect an approximate doubling of the number of detectable red neurons 1872 ± 623 (means \pm SEM) in R-AC at Day 21 as shown in Reply Fig. 4a.

Reply Fig. 4: Immunostaining increases the sensitivity for measuring the number of H129_{Amp} labeled neurons. Elements of this can be found in the revised Fig. 2 and 3 in the manuscript.

7. There is no information about the amount of the virus injected into the brain and the stereotaxis coordinates.

Reply: We apologize for not making it clear. In the revised manuscript, we have added all necessary experimental parameters, including tracer/helper doses, injection stereotaxis coordinates, animal species, and observation time points, to the corresponding figure legends.

8. The authors also indicated that they had determined empirically that after 7 days, the number of the potential co-labeled (potential starter) decreased and dimmed. Can the authors further explain this? Are the neurons dead?

Reply: The reviewer accurately points out the most important disadvantage of the tracer system even at its present state-of-the-art. Unfortunately, toxicity of the H129_{Amp} tracer system remains relatively strong in starter neurons. The starter neurons are coinfecting by the tracer and helper, which act cooperatively and support each other's replication. This leads to high levels of viral protein expression and strong toxicity in the starter neurons. At day 1, most potential starter (coinfecting) neurons display normal morphology (Supplementary Fig. 4b). By day 5 post injection, the morphology of many neurons is abnormal and show rounded morphology (Supplementary Fig. 4f). By day 7, the signal of the potential starter dramatically decreases and dims (Supplementary Fig. 4h). Based on available evidence, we believe that the replication of the viruses (tracer and helper together) is the primary cause for death of the starter neurons.

9. The authors stated that: " Many GFP+ neurons were clearly observed in downstream nuclei are directly innervated by AC neurons, including contralateral auditory cortex (Cont. AC), lateral amygdaloid nucleus (LA), medial geniculate nucleus (MG), external globus pallidus (GPe) and locus coeruleus (LC) (Fig. 2e-i), but not in further downstream second-order connected nuclei (data not shown). These results show that H129Amp tracer anterogradely labels directly connected postsynaptic neurons limited to monosynaptic connections." These results are not showing that it is due to monosynaptic connections.

Reply: We agree that the results shown in Fig. 2 alone cannot sufficiently conclude that the

labeling observed is due entirely to monosynaptic connections. However, by combining the tracing mechanism and other experimental observations based on our careful optimization of the tracer system, we can draw the conclusion that the H129_{Amp} tracer system anterograde transmits monosynaptically.

i) The tracing mechanism indicates the monosynaptic transmission of the H129_{Amp} tracer system. As we introduced in the manuscript, the H129_{Amp} tracer and helper mutually support each other's replication. But the Cre expressed by H129_{Amp} tracer excises the LoxN-flanked *pac* from the helper genome, and disarms the genome packaging ability to prevent helper production. H129_{Amp} is packaged into a complete viral particle and transmits to the postsynaptic neuron, and is then restricted there due to the lack of helper. Therefore, the H129_{Amp} tracer may perform monosynaptic anterograde tracing.

ii) Tracer or helper alone don't spread. We have shown this by careful examination of the injection site and its representative downstream regions when applying the tracer or helper alone at different time points (Supplementary Fig. 2 and 3). No labeled neurons are observed following injection of either tracer or helper alone, indicating the H129_{Amp} tracer system achieves tracing only in combination with both the tracer and the helper.

iii) No multisynaptic tracing is observed. As we show in Reply Fig. 2, no helper-labeled postsynaptic neurons are detected even at very early time points (Days 1-3) of the tracing. This result, in combination with the helper alone tracing results introduced in ii) (Supplementary Fig. 3), shows that the helper alone doesn't label postsynaptic neurons, and therefore does not lead to uncontrolled multi-synaptic transmission of the tracer. Altogether, we conclude the results shown in Fig. 2 are due to the anterograde monosynaptic tracing of the H129_{Amp} tracer system. We have revised the text related to Fig. 2 in the revised manuscript to reflect these points (lines 238-247).

10. Can the authors provide more information about the *Rph3a-Cre* transgenic mice (i.e., cell type, layers, etc.)?

Reply: We concur that this should be better explained. Rph3A is known as a synaptic vesicle-associated protein involved in the regulation of exo- and endocytosis processes at presynaptic sites. According to the Rph3A ISH of Allen Reference Atlas (Experiment #71587899), Rhp3A positive neurons are largely distributed in Layer 2 to Layer 6 in the cortex, and a small amount of Rhp3A neurons are present in Layer 1. This Rph3A-Cre transgenic mouse strain was created by inserting P2A-iCre between Exon 22 and 3'-UTR region of Rph3a (Beijing Biocytogen Co., Beijing, China). We are currently using this line for other research, and there is no citable publication so far. We now add additional explanatory text concerning this to the revised manuscript (lines 259-260).

11. The authors stated:" Next, we mapped the outputs of the input-defined neuron subpopulation anatomically and functionally with H129Amp tracer system. H129Amp tracer system (H129Amp-CTG tracer and helper) was administrated into the AC in the left hemisphere (L-AC) of wildtype C57BL/6 mice as described above, and AAV2/9-DIO-ChR2-mCh was simultaneously injected into the right hemisphere AC (R-AC) in the same mice (Fig. 3a). R-AC receives inputs from the L-AC, and projects back to L-AC, as well as the right hemisphere MG and LA (R-MG and R-LA) (Fig. 3a). Newly propagated H129Amp

tracer transmits from the L-AC to the postsynaptic neurons in the R-AC and provides Cre allowing the AAV2/9-DIO-ChR2-mCh to express ChR2-mCherry. As we expected, on Day21 the soma of mCherry labeled neurons (mCh+) were clearly observed in the R-AC (Fig. 3b), but not in any other regions (data not shown)." (1) Can the authors provide a better image of the R-AC (figure 3b)? It is impossible from the actual image concluding that it is the R-AC. (2) What is the fraction of neurons that are positive to ChR2? (3)It looks like that there are more ChR2-positive (figure 3b) than GFP-positive (figure 2h) neurons in the contralateral AC. Why?

Reply (1): We concur that this can be presented more clearly. We have replaced the R-AC depiction (revised Fig. 3b) to now show the coronal right hemisphere field to better display the R-AC.

Reply (2): ChR2 and mCherry are expressed simultaneously as a fusion protein, mCherry at the C terminal end of ChR2 as a fusion protein, with expression directed by AAV2/9-DIO-ChR2-mCh in the presence of Cre-recombinase. Therefore, all neurons labeled by mCherry are ChR2 positive. We counted mCherry+ (mCh+) neurons in the R-AC at Day 21, and find an average of 1872 ± 623 (mean \pm SEM) red neurons (Reply Figure. 4a).

Reply (3): We concur with the reviewer that this is an important issue. Following the reviewer comments, we performed quantitative analysis of the number of labeled neurons. There are an average of 740 ± 146 (means \pm SEM) green neurons (GFP+) were observed in Cont. AC at Day 5 in Fig. 2. While 1872 ± 623 (means \pm SEM) red neurons (mCh+), over 2-fold higher than GFP+ neurons, were detected in R-AC at Day 21 in Fig. 3. The difference is most likely due to the signal amplification effect of the Cre-Lox reporter system that is applied for the experiments shown in Figure 3, as detailed for our reply to Comments #6 of Reviewer #3 described above. All confocal microscopy imaging is limited to fluorescence signal that can be measured above the detection threshold, suggesting that fluorescent proteins must accumulate to a detectable level. Therefore, the observed GFP+ neurons don't represent all neurons labeled by transmitted H129_{Amp}, since the GFP protein level in some neurons was not high enough for detection directly. We've confirmed this by anti-GFP immunostaining, which increases the detectable neuron number due to the higher sensitivity of this method (Reply Fig. 4b and c). The Cre/lox dependent reporter system applied in Fig. 3 contributes to signal amplification, and also results in more detected neurons. H129_{Amp}-CTG tracer anterogradely transmitted to R-AC, and expresses Cre recombinase in those neurons, which in turn triggers ChR2-mCh expression from the AAV2/9-DIO-ChR2-mCh. Cre-recombinase sufficiently catalyzes the recombination of AAV2/9-DIO-ChR2-mCh genome, and permanently turns it into a constitutive expression form, thus leading to the 3 week accumulation of ChR and mCh and the greater number of cells detected relative to the number of cells observed by GFP labeling as noted above.

Reply Fig. 4: Immunostaining increases the sensitivity for measuring the number of H129_{Amp} labeled neurons. Elements of this can be found in the revised Fig. 2 and 3 in the manuscript.

12. All the Experiment performed in figure 3 and 4 need further experiments before a conclusion like this can be drawn:" Altogether, these results demonstrate that the H129Amp tracer systems (H129Amp-CTG and H129Amp-Flp-DIO-TG tracers) in combination with the appropriate reporter AAVs are capable to not only anatomically but also functionally map the output pathways of subpopulations of input-defined neurons."

Reply: Compared to other H129 tracers, H129_{Amp} tracer system displays little or no toxicity in the postsynaptic neurons. This enables ChR2-assisted optogenetic assay (please see the revised Fig. 3). Based on these feature, it indicates the H129_{Amp} tracer system may potentially contribute to the functional connection mapping of the input-defined neurons. And in Fig. 4, we achieved anatomically mapping the outputs of input-defined subpopulation neurons by combination application of H129_{Amp} tracer system and H129-dgK tracing system. This suggests the H129_{Amp} tracer system can be used to anatomically map the input-defined functional outputs.

However, we do agree with the reviewer that these experiments can only demonstrate the potential of the H129_{Amp} tracer system, but are not sufficiently developed yet to fully support our earlier conclusion. Taking all of this into account, we have rephased this in the revised manuscript as "Altogether, these results demonstrate that the H129_{Amp} tracer systems combined with appropriate reporter AAVs may be used to map the output pathways of input-defined neuron subpopulations both anatomically and functionally (ChR2-assisted)"(lines 319-321).

In summary, we thank the reviewers for their constructive suggestions and the editors for the opportunity to address the reviewers concerns.

Reviewers' Comments:

Reviewer #2:

Remarks to the Author:

The authors have done an excellent job of responding to the previous review. I am satisfied with all their responses to the technical issues and thank them for their serious consideration and the addition of new data and experiments to address the issues that were raised. But none of the new experiments have been described in the Results section of the revised manuscript. I believe that this should be remedied so that readers will know about the potential caveats that they need to address with adequate controls when they use the system and so that they will have confidence that the potential artifacts can be avoided. In particular, the following results should be described in the main text, even if the figures supporting them are supplementary.

- 1) The lack of direct retrograde in infection with either the helper or the tracer. Describe why the controls are needed and describe the result. (Supplementary Figures 2-3.)
- 2) The lack of labeling outside the local cortical area when the Cre-dependent tracer (H129Amp-DIO-TG) and helper are injected into the cortex of PV-Cre mice. (Supplementary Fig. 13.)

When describing the results from PV-Cre mice in the main manuscript it should also be made clear in the main text that these results were achieved with particular titers of the tracer and helper and the results should make reference to the details that are further described in the supplemental material.

The introduction states:

"To date, there are five published anterograde monosynaptic tracer systems which belong to three types of viruses."

Please consider the following anterograde transsynaptic tracing method in the Introduction.

1. Tsai NY, Wang F, Toma K, Yin C, Takatoh J, Pai EL, Wu K, Matcham AC, Yin L, Dang EJ, Marciano DK, Rubenstein JL, Wang F, Ullian EM, Duan X: Trans-Seq maps a selective mammalian retinotectal synapse instructed by Nephronectin. *Nat Neurosci* 2022, 25:659-674.

Point-by-point responses to the Reviewers' comments.

Reviewer #2 (Remarks to the Author):

The authors have done an excellent job of responding to the previous review. I am satisfied with all their responses to the technical issues and thank them for their serious consideration and the addition of new data and experiments to address the issues that were raised. But none of the new experiments have been described in the Results section of the revised manuscript. I believe that this should be remedied so that readers will know about the potential caveats that they need to address with adequate controls when they use the system and so that they will have confidence that the potential artifacts can be avoided. In particular, the following results should be described in the main text, even if the figures supporting them are supplementary.

Reply: We thank the Reviewer for their acknowledgment of the novel tracing system we describe here in our manuscript, for their positive feedback on our revision, and for their constructive comments that have helped us to strengthen our article. We have accordingly revised the manuscript and we now describe the new experiments in the Results section for new and unpublished results that we have added so that readers will have a clear view of this novel tracing system for the purposes of interpretation and their own implementation.

1) The lack of direct retrograde in infection with either the helper or the tracer. Describe why the controls are needed and describe the result. (Supplementary Figures 2-3.)

Reply: We thank the Reviewer for pointing this out. We now include the requested control experiments that are described in detail in the revised manuscript (lines 258-265).

2) The lack of labeling outside the local cortical area when the Cre-dependent tracer (H129Amp-DIO-TG) and helper are injected into the cortex of PV-Cre mice. (Supplementary Fig. 13.)

When describing the results from PV-Cre mice in the main manuscript it should also be made clear in the main text that these results were achieved with particular titers of the tracer and helper and the results should make reference to the details that are further described in the supplemental material.

Reply: We concur and have added these details in the revised manuscript (lines 507-519) and figure legend of Supplementary Fig. 11.

The introduction states:

“To date, there are five published anterograde monosynaptic tracer systems which belong to three types of viruses.”

Please consider the following anterograde transsynaptic tracing method in the Introduction.

- 1. Tsai NY, Wang F, Toma K, Yin C, Takatoh J, Pai EL, Wu K, Matcham AC, Yin L, Dang EJ, Marciano DK, Rubenstein JL, Wang F, Ullian EM, Duan X: Trans-Seq maps a selective mammalian retinotectal synapse instructed by Nephronectin. Nat Neurosci 2022, 25:659-674.*

Reply: We appologized for missing this recently reported tracer, and thank the reviewer for bringing this to our attention. We now include this citation and have amended the revised manuscript (lines 101-120).